# Understanding the learned look-ahead behavior of chess neural networks

## Abstract

We investigate the look-ahead capabilities of chess-playing neural networks, specifically focusing on the Leela Chess Zero policy network. We build on the work of Jenner et al. (2024) by analyzing the model's ability to consider future moves and alternative sequences beyond the immediate next move. Our findings reveal that the network's look-ahead behavior is highly context-dependent, varying significantly based on the specific chess position. We demonstrate that the model can process information about board states up to seven moves ahead, utilizing similar internal mechanisms across different future time steps. Additionally, we provide evidence that the network considers multiple possible move sequences rather than focusing on a single line of play. These results offer new insights into the emergence of sophisticated look-ahead capabilities in neural networks trained on strategic tasks, contributing to our understanding of AI reasoning in complex domains. Our work also showcases the effectiveness of interpretability techniques in uncovering cognitive-like processes in artificial intelligence systems.

## 1. Introduction

Recent advances in artificial intelligence have produced systems capable of superhuman performance in complex domains like chess and Go (Silver et al., 2018). However, the mechanisms underlying these systems' decision-making processes remain poorly understood. A key question is whether neural networks trained on such tasks learn to implement sophisticated planning algorithms, or if they rely primarily on pattern matching and heuristics.

This paper builds on recent work by Jenner et al. (2024) that found evidence of learned look-ahead behavior in a chess-playing neural network. We extend their analysis to examine

---

[1]Anonymous Institution, Anonymous City, Anonymous Region, Anonymous Country. Correspondence to: Anonymous Author <anon.email@domain.com>.

Preliminary work. Under review by the International Conference on Machine Learning (ICML). Do not distribute.

longer-term planning capabilities and the consideration of alternative moves. Specifically, we investigate whether the network encodes information about future board states and potential move sequences beyond just the next move.

Understanding the internal reasoning processes of these models is important for several reasons. First, it provides insights into the nature of intelligence that emerges from neural network training, potentially informing our understanding of both artificial and biological cognition (McGrath et al., 2022). Second, it has practical implications for improving AI systems in strategic domains, as a deeper understanding of their planning mechanisms could lead to more efficient and robust architectures (Czech et al., 2024). Finally, it contributes to the broader field of AI interpretability, which is essential for building trustworthy and controllable AI systems (Chattopadhyay et al., 2019).

In this context, understanding the depth and sophistication of learned look-ahead behavior is particularly relevant. While Jenner et al. (2024) has demonstrated the existence of look-ahead behavior in chess models, understanding how this capability scales to longer sequences is important for several reasons. First, it helps us understand the limits of learned look-ahead behavior - whether models can truly chain together long sequences of moves or if they rely primarily on short-term patterns. Second, analyzing how the model processes moves at different time horizons can reveal whether it uses similar or different mechanisms for near-term versus long-term planning. Finally, understanding these capabilities in chess provides insights that may generalize to other domains where long-term planning is essential, such as robotics or strategic decision-making.

Recent work in mechanistic interpretability has made significant strides in understanding the internal workings of language models (Geva et al., 2023; Wang et al., 2023) and game-playing models (Li et al., 2023; Nanda et al., 2023). However, most of these studies have focused on relatively simple tasks or isolated components of larger systems. Our work aims to bridge this gap by analyzing sophisticated planning behavior in a state-of-the-art chess engine.

Chess provides an ideal testbed for this study due to its well-defined rules, clear strategic elements, and the availability of strong neural network-based models (Ruoss et al., 2024). Unlike language models, where the notion of "cor-

rect" behavior is often ambiguous, chess allows for precise evaluation of model performance and decision-making. The game's complexity requires long-term planning and consideration of multiple possible futures, making it a rich domain for studying advanced cognitive processes in AI systems.

Our key contributions are:

- Showing that **the model's look-ahead behavior is highly dependent on the specific type of chess position**, with different piece capture and checkmate scenarios being stored differently in the residual stream, and processed differently by multiple attention heads;

- Extending the analysis of Jenner et al. (2024) of look-ahead behavior to the 5th and 7th future moves in chess positions. Specific attention heads seem strongly responsive to longer term future moves, and **the model appears to process some future moves using similar concrete internal mechanisms**;

- Demonstrating that **the model considers multiple move sequences, not just a single line of play**. Moreover, corrupting the board squares relevant to one move sequence often leads the model to pick the alternative move sequence, as expected for look-ahead behavior.

These findings provide new insights into the look-ahead capabilities that can emerge in neural networks trained on strategic planning tasks. They also demonstrate how interpretability techniques can uncover sophisticated cognitive processes in AI systems. Our work contributes to the growing body of research on AI planning and reasoning (Chen et al., 2021; Hao et al., 2023; Ivanitskiy et al., 2023; Garriga-Alonso et al., 2024), offering a detailed look at how these capabilities manifest in a complex, real-world domain.

To obtain these results, we construct novel approaches to analyze the model's look-ahead behavior, extending the techniques used in Jenner et al. (2024). We introduce a puzzle set notation that disentangles the model's behavior for different types of chess positions, and enables a clearer analysis of the model's look-ahead behavior for higher move counts. We use activation patching to measure the causal importance of different board squares in the model's decision-making process, probing to test the prediction accuracy of the model's future moves, and ablation to identify the attention heads that are responsible for the model's look-ahead behavior. By showcasing how these techniques can be used in a complementary manner, we expect their usefulness to extend to future mechanistic interpretability studies of other models with potential look-ahead or planning capabilities.

We also adapt the board corruption technique used in Jenner et al. (2024) to work for multiple move sequences, and apply it to analyze the model's consideration of alternative moves. This analysis should be suitable for future studies of planning behavior in other domains, by making it easier to produce contrastive pairs for activation patching, thereby enabling a more fine-grained analysis of the model's behavior for different look-ahead strategies.

## 2. Setup

This section describes the chess model, dataset, analysis techniques, and notation used in our analysis. All experiments were run using an RTX 3070Ti, with a combined runtime of 2 days. For reproducibility, additional details are available in Appendix H.

### 2.1. Chess Model

In this study, we analyze the Leela Chess Zero (Leela) policy network, which is part of a larger MCTS-based chess engine similar to AlphaZero (Silver et al., 2018). Leela is currently the strongest neural network-based chess engine (Haworth and Hernandez, 2021). Its policy network takes a single board state as input and outputs a probability distribution over all legal moves.

Leela is a transformer that treats each of the 64 chessboard squares as one sequence position, analogous to a token in a language model. This architecture allows us to analyze activations and attention patterns on specific squares. The version of Leela we use has 15 layers and 109 million parameters. Due to peculiarities of this particular model, explained in Jenner et al. (2024), we use a finetuned version of the model, trained and used by Jenner et al. (2024).

### 2.2. Dataset

We use Lichess' 4 million puzzle database as a starting point. Each puzzle in our dataset has a starting state with a single winning move for the player whose turn it is, along with an annotated principal variation (the optimal sequence of moves for both players from the starting state).

In our analysis, we refer to moves in the principal variation as follows: The **1st move** is the initial move made by the player in the starting position. The **2nd move** is the opponent's response to the 1st move. The **3rd move** is the player's follow-up move after the opponent's response. We extend this notation to refer to subsequent moves (e.g., 5th move, 7th move) when analyzing longer sequences.

The puzzles were curated into three datasets: a 22k puzzle dataset used in Jenner et al. (2024), solvable for the Leela model but difficult for weaker models to solve, and used for the 3 and 5-move analysis; a 2.2k dataset of 7-move puzzles; and 609 puzzles for the alternative move analysis. Additional details on the dataset generation can be found in Appendix H.

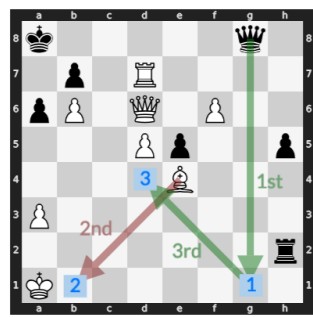

Figure 1: Examples of 3-move puzzles in puzzle set 112 (left) and 123 (right). "1st", "2nd", and "3rd" mark the move order, with the green (resp. red) arrow indicating the optimal move of the player (resp. opponent). The board squares the piece moves to are marked in blue. They are listed sequentially starting from 1. The resulting number sequence labels the associated puzzle set, with *1st move ↦ square 1, 2nd ↦ sq. 1, 3rd ↦ sq. 2* resulting in the set 112, for example. For these two examples, the optimal move sequence (i.e. principal variation) results in a checkmate, which may be marked with the prefix M, so they additionally belong to the subsets M112 and M123, respectively.

## 2.3. Analysis Techniques

We employ three main techniques to analyze the internal representations of the model:

**Activation Patching.** This technique, also known as causal tracing (Meng et al., 2023), allows us to measure the causal importance of specific model components. For a given board state and model component (such as a particular square in a particular layer), we replace the clean activations of that component with those from a different "corrupted" board state. If this intervention significantly changes the model's output, it suggests that the patched component contained necessary information about the clean state that differed in the corrupted state. In our chess setup, we employ the approach of Jenner et al. (2024), where the corrupted board state is a minimally modified version of the original board state, where the optimal next move is different but still non-trivial (see Appendix H for more details). Patching then consists of replacing the clean activations (from a particular layer or attention head) associated with a particular board square by their corrupted counterparts, generated using the corrupted board as input to the model.

**Probing.** We use linear probes to decode information from the model's internal representations. A probe is a small classifier trained to predict certain information (e.g., the position of a piece or a future move) from the model's hidden states.

High probe accuracy suggests that the probed information is explicitly encoded in the model's representations. In our setup, we use probes to test the prediction accuracy for the puzzles' future moves, based on the model's internal states when given the current board state as input (see Figure 3).

**Ablation.** We employ zero ablation, particularly when analyzing attention heads. In this technique, we selectively set certain weights or activations to zero, effectively removing their contribution to the model's output. By comparing the model's performance before and after ablation, we can assess the importance of specific components (such as individual attention heads or attention patterns) to the model's decision-making process. This method is useful for identifying key mechanisms involved in look-ahead behavior. In our setup, we apply zero ablation to individual weights in specific attention heads, in order to determine which board squares certain attention heads are mainly attending to.

These techniques allow us to investigate how the model represents and processes information about current and future board states, providing complementary insights into its look-ahead capabilities. While activation patching reveals what information is causally necessary for the model's decisions, probing can identify information that is encoded but not necessarily used for the final move choice. For example, our probing results show that the model encodes information about opponent moves, even when patching does not provide conclusive causal evidence. Similarly, while patching provides a broad causal view applied across the entire model, ablation provides a fine-grained view of which board squares the model's attention heads are attending to, giving us a better qualitative understanding of the model's behavior.

### 2.4. Puzzle set notation

In (Jenner et al., 2024), it was observed that the Leela model internally treats cases where the player's moved piece is immediately captured by the opponent differently from cases where the opponent piece moves to an unrelated square (see Figure 1). When considering more complex future move sequences, the increasing number of different scenarios treated distinctly by the model makes its analysis challenging. To combat this problem, and disentangle the model's behavior for different cases, we introduce a new labelling approach for each chess puzzle that we analyze.

We start by separating the data into different puzzle sets depending on the similarity between the board squares involved. In particular, for each player and opponent move, we label the move based on the square the piece moves to. For the analysis, we do not consider the squares the pieces start in, as we verify that this additional complexity does not play a significant role in the model's internal behavior (see Appendix A), as previously observed in Jenner et al. (2024).

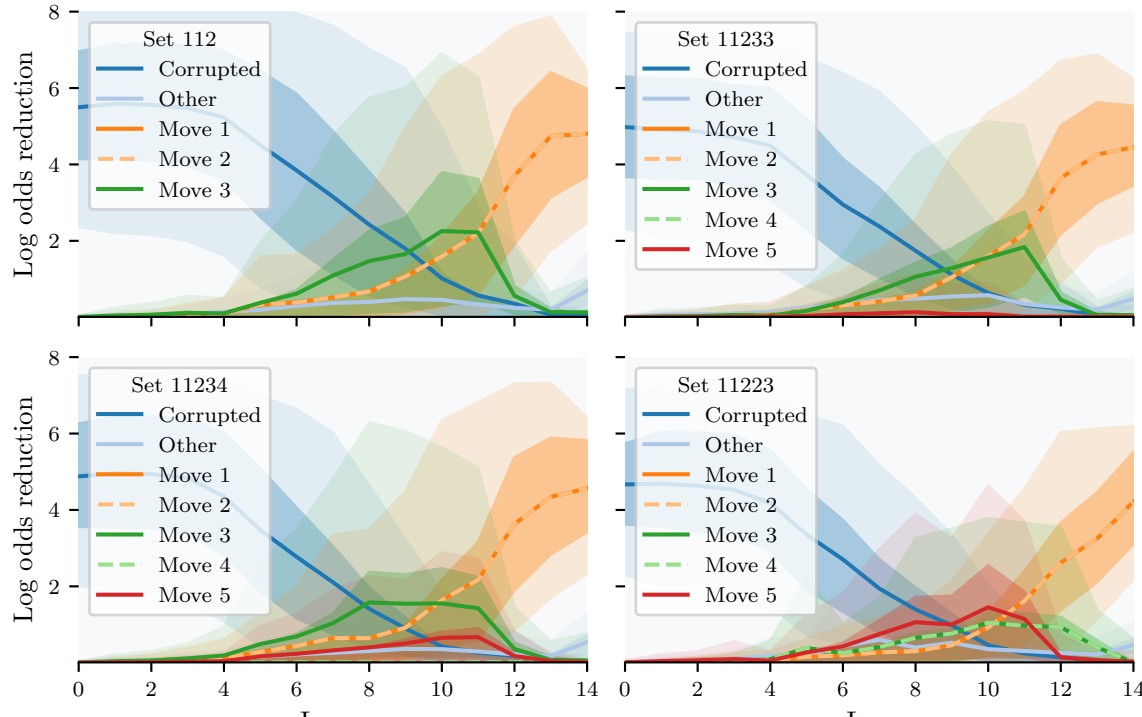

Figure 2: Log odds reduction of the correct move as a result of activation patching, for 5-move puzzle sets of the form 112XY, where $Y > 2$ (i.e. the fifth move square is distinct from the first and third move squares). "Corrupted" indicates the patched square from the corrupted board. The label $i$ indicates the move square for the $i$-th move, with solid (resp. dashed) lines indicating the destination square for the player (resp. opponent) piece. "Other" indicates the contributions of the remaining squares. Dashed lines indicate opponent moves. Confidence intervals of 50% and 90% are displayed using darker and lighter hues, respectively, indicating the distribution of the log odds reduction accross the puzzles considered.

Since there are far more combinations when considering a larger number of moves, we use the notation $s_1 s_2 \cdots s_n$ to refer to a sequence of squares, where $s_i = s_j$ iff the $i$-th and $j$-th move squares are the same. Since we are considering up to 7 moves, all $s_i$ are one single digit. We start with $s_1 = 1$ and raise the digit used whenever a new square is different from the ones in previous moves. As shorthand, we may use uppercase letters to represent arbitrary digits. Starting alphabet letters (like A, B, and C) are used to represent distinct digits, while ending alphabet letters (like X, Y, and Z) are used to represent any digit combination. For instance, while the notation 111XY = {11111, 11112, 11122, 11123} would represent any puzzle set starting with 111, the notation 111AB = {11112, 11123} represents the 2 puzzle sets starting with 111 where the final two squares are distinct. The set {11111, 11122} could be represented by both approaches, using either 111AA or 111XX. Additionally, we occasionally may prefix the sequence with the letter M (resp. N), to denote the subset of puzzles where the optimal move sequence results (resp. does not result) in a checkmate.

Using this notation, we would represent the 3-move scenarios considered in Jenner et al. (2024) as 112 (resp. 123) for

the puzzle sets where the 1st and 2nd move squares were the same (resp. different), and the 1st and 3rd move squares were distinct. See Figure 1 for examples of the notation.

Puzzles with more than 3 moves are also included in Jenner et al. (2024), but its analysis bundles the higher move squares (fifth, seventh, etc.) into the third move square results, which makes it difficult to see if the model is able to concretely look ahead past the third move. In our analysis, we disentangle the results for each puzzle length.

## 3. Results

In this section, we verify that the Leela chess model looks ahead into the fifth and seventh moves when solving chess puzzles, and later shown evidence that the model is able to consider multiple future branches when choosing the best move to play.

**The starting move squares do not play a significant direct role.** While the starting move squares (i.e. the squares the pieces start in before they are moved) are generally critical for assessing the right next moves for each player, the results

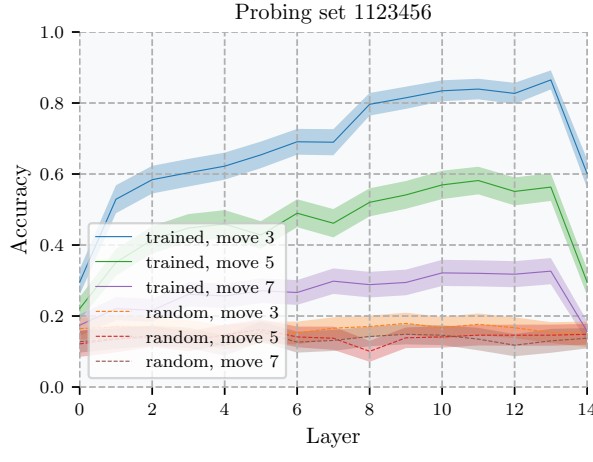

Figure 3: Probing the model's residual stream for the puzzle set 1123456. The probe's accuracy decreases as we look into more distant future move squares, with the 7th move square's accuracy being considerably low, but still non-negligible when compared with the probe's accuracy for a random model. The observed accuracy increases as we traverse the model's layers, as the residual stream contains the move information in a way that is progressively easier to decode. The sharp dropoff at the last layer likely stems from the model's lack of use of future move information by the policy and value heads, instead relying more strongly on the next move information (see Appendix C).

obtained in our analysis are consistent with the conclusion in Jenner et al. (2024) that the starting move squares do not seem to play a significant *direct* role in the look-ahead behavior of the model. Instead, the model seems to process the board state by directly encoding the moves of interest in the associated squares the pieces move to. Consequently, the model responds strongly to corruptions on the destination squares of moves, while showing negligible effects for the starting squares (see results in Appendix A). Therefore, we only focus our analysis on the squares the pieces move to during each move, and ignore the squares the pieces start in.

**The model considers up to the seventh future move when choosing the best next move.** We show probing results for the puzzle set 1123456 in Figure 3, and additional results in Appendix C. The probe's accuracy decreases as the move square becomes increasingly more distant from the present, with the 7th move square's accuracy being considerably low, but still non-negligible. Activation patching also show higher future move squares playing an important role in the model's performance (see Figure 2 and Appendix B).

**The model behavior is highly dependent on the puzzle set.** The results of patching the model's residual stream

for some 3 and 5-move sets are presented in Figure 2, with additional results shown in Appendix B. Only sets with more than 50 puzzles are considered. We note that patching the fifth move square has a non-negligible effect on the log odds of the correct move for most 5-move puzzle sets. The effect is most salient for the set 11223, while not being very significant for set 11233.

The results of patching the attention heads for 3-move puzzle sets can be seen in Figure 4, with higher move sets in Appendix E. We note marked differences between the sets, with the L12H12 attention head (i.e. head 12 in layer 12) being the most important for the set 112, but playing a weaker role in the remaining sets. Moreover, the set 111 seems to respond more strongly to attention heads L11H10 and L11H13, which do not seem to play a significant role in the other sets. Sets 122 and 123 do not respond strongly to patching any of the attention heads. Additionally, in Figure 5, we observe that the behavior of some attention heads varies notably depending on whether or not the board position will soon result in a checkmate, indicating that **the model behavior is also dependent on the near-term possibility of checkmate**. Additional results can be found in Appendix E.

Overall, we note that the importance of the future move squares is highly dependent on the puzzle set, suggesting that the Leela model does not treat the sets similarly. Further corroborating results can be found in Appendix B.

In no puzzle set does a distinct second or fourth move play a significant direct role. Nonetheless, probing results (see Appendix C) suggest that the model does contain information about the second and fourth move squares, but possibly in an indirect way that is not straightforwardly captured by patching techniques.

**The model processes 3rd, 5th, and 7th moves similarly.** In Figure 2, we note that the saliency of the fifth move square varies significantly between the sets, with the set 11233 barely displaying an effect, followed by 11234, and with the set 11223 displaying a strong effect. Based on Figure 2 and Appendices B and E, we hypothesize that the model may be using similar mechanisms to consider the 3rd, 5th, and 7th moves. We particular, we note that patching shows weak, moderate, and strong effects for move C for puzzle sets of the form (· · · )ACC, (· · · )ABC, and (· · · )AAC, respectively (where ellipsis stands for arbitrary preceding moves). The corresponding puzzle sets shown in Figure 2 for these three cases are puzzle sets 11233, 11234, and 11223, respectively. The 7th move appears to be near the model's look-ahead limit.

We note from Figure 4 (and Figures 24 to 26 in Appendix E) that some heads matter a lot for the fifth and seventh move analysis.

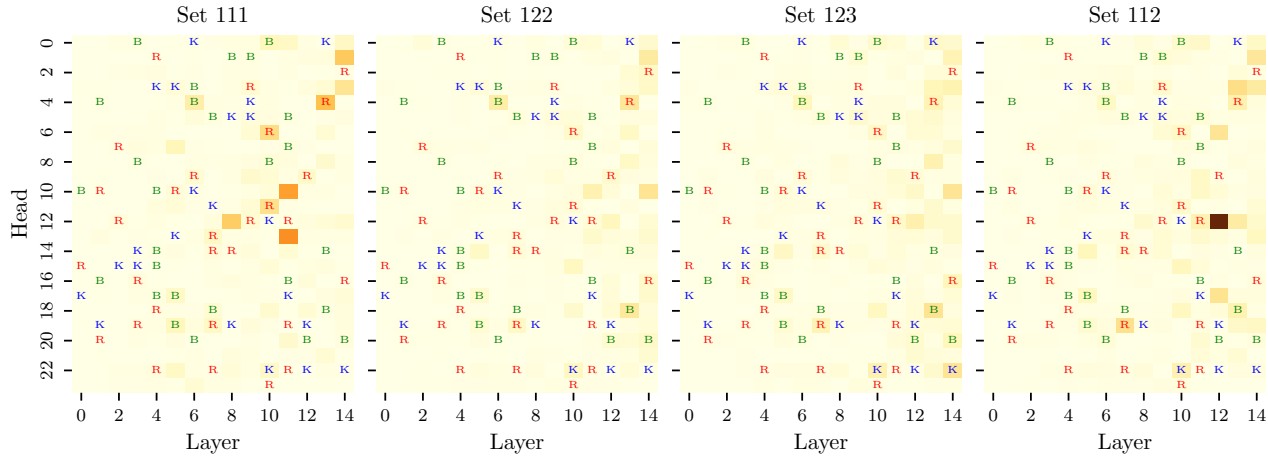

Figure 4: Attention head patching for puzzles with 3 moves. Darker tones indicate higher log odds reduction of the correct move. The letters K, B, and R represent the king, bishop, and rook attention heads, respectively, identified in Jenner et al. (2024). Darker colors mark a higher log odds reduction due to patching, with the highest being 0.73, for L12H12 (head 12 in layer 12) in set 112.

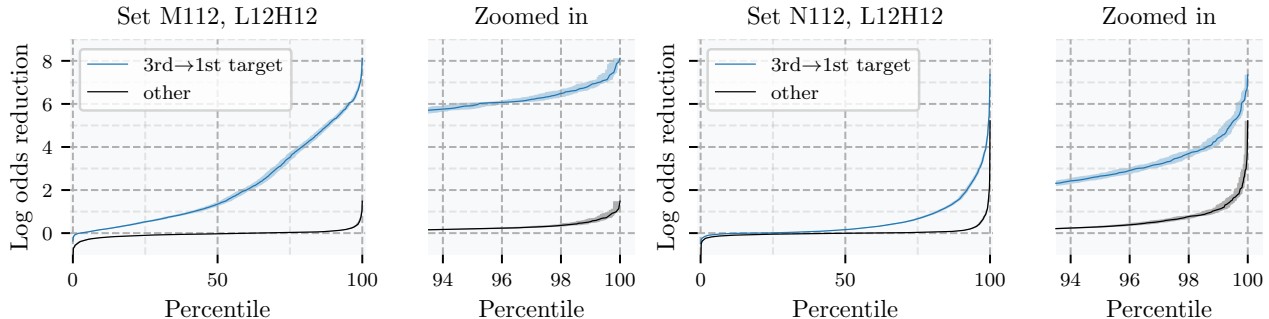

Figure 5: Ablation results of the L12H12 head for checkmate (M112, left) and non-checkmate (N112, right) puzzle set 112. We note that head L12H12 not only appears to mainly move information "backward in time", i.e. from the third to the first move square, but it appears to be especially critical in scenarios that explicitly result in a checkmate (in this case, in 3 moves).

**L12H12 is also important for 5th and 7th moves.** In Jenner et al. (2024), it was shown that attention head L12H12 moves information "backward in time" from the third to the first move square, for some 3-move puzzles.

In Figure 5, we show that L12H12 is also important for the 5th and 7th moves. The results are shown in Figures 16 to 18 in Appendix D. We note that, for puzzles with five moves, L12H12 may be responsible not only for moving information backward in time from the third to the first move square, but also from the fifth move square.

We hypothesize that the attention head moves information backward in time from square C to A (or the 1st move square) when the puzzle set has the form $(\cdots)AAC(\cdots)$, and to a lesser extent when it has the form $(\cdots)ABC(\cdots)$, while not responding to the form $(\cdots)ACC(\cdots)$. When the set matches the pattern at multiple turns, the later turn often takes precedence (for instance, we would expect 11223, which matches both $AAC(\cdots)$ and $(\cdots)AAC$, to mainly move information from the 5th, and not from the 3rd move square). Its behavior mimics that seen from the general activation patching results, discussed in more depth in Appendix B. Overall, the specific patterns that L12H12 responds to appear to be time-insensitive - that is, the patterns AAC, ABC, and ACC may apply to moves 1-2-3, moves 3-4-5, or moves 5-6-7. This suggests the model has learned some general pattern-matching mechanisms across time rather than timing-specific heuristics.

We also note that L12H12 strongly responds to moves that may result in checkmate, as previously seen in Figure 5. We

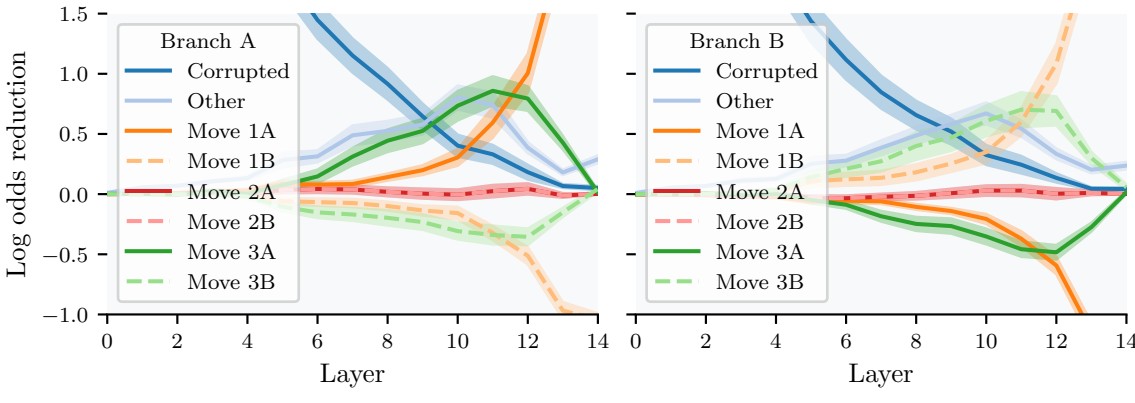

Figure 6: Patching results of the alternative move analysis, for puzzle set 123425 (where the last 3 digits stand for the alternative branch squares). The log odds reduction for the next move for branch A (left) and branch B (right) are shown. Negative log odds reduction for branch A (resp. B) implies that patching the square improves the model's odds of choosing the main (resp. alternative) move branch. Solid (resp. dashed) lines indicate the main (resp. alternative) move squares. Shaded regions mark the standard deviation for the mean of the log odds reduction accross the puzzles considered.

further investigate this behavior in various checkmate scenarios, including puzzles with multiple checkmate options, as detailed in Appendices D and G. The model generally appears to have checkmate-specific mechanisms, which are not triggered for non-checkmate scenarios.

**Other heads also play a crucial role for complex puzzles.** We perform the same detailed analysis for the L12H17, L13H3, L11H13, and L11H10 heads, which showed the highest log odds reduction after L12H12 (see Figure 4 and Appendix E). The results and discussion are shown in Appendix D. Our analysis reveals distinct roles for these attention heads.

Head L12H17 appears to move information "backward in time" for puzzle sets of the form AABCD, where C is different from D, and D is preferably equal to A. Notably, in sets of the form AABCA, the model relies more heavily on L12H17 than on L12H12. This head also only seems to play a major role in longer scenarios, as the performance downgrade from patching is not significant for 3-move puzzles. Interestingly, unlike L12H12 (see Figures 5 and 22), head L12H17 appears to respond more strongly to puzzle sets that do not result in checkmate (see Figure 23). It may possibly complement head L12H12 in moving information backward in time.

Attention head L13H3 seems to move information "backward in time" for puzzle sets of the form AABCD, where either C=D or B=C. However, its role is less pronounced compared to L12H12 and L12H17. The roles of L11H10 and L11H13 are less clear based on the ablation results alone. While some puzzle sets show responses to these heads in the attention patching analysis, the ablation results suggest their contributions may be more subtle or indirect.

Interestingly, our analysis of checkmate vs. non-checkmate scenarios reveals that L12H12 plays a more significant role in moving information backward in time in checkmate scenarios, while L12H17 is more active in non-checkmate scenarios. This differentiation suggests that the model may process checkmate and non-checkmate positions using distinct mechanisms, highlighting the context-dependent nature of its information processing strategies. The emergence of these specialized components through training, without explicit programming, demonstrates how neural networks can develop sophisticated information processing strategies for planning tasks. This provides valuable insights into how models might learn to handle complex sequential decision-making in other domains.

**The model considers alternative move sequences.** We investigate to what extent the model considers alternative moves, focusing on situations where there are two relatively equally good moves to play, which we label as the main move branch A and the alternative move branch B. To simplify the analysis, for this section, we restrict our attention to 3-move puzzles where the Leela model assigns a probability around $1/2$ of choosing each of the two move branches. We consider puzzles with two branching sets of moves, each with distinct first and third move squares (for a total of 4 distinct squares). For activation patching, we consider corrupted boards which are compatible with both branches A and B. See Figure 29 for examples and Appendix F for details.

We show some results in Figure 6, with full results in Appendix F. Patching the alternative first move square (1B) consistently has a strong positive effect on increasing the model's odds of choosing the main first move (1A), and vice-versa, demonstrating the model's ability to weigh im-

mediate alternatives. Patching the alternative *third* move square (3B) often improves the model's odds of choosing the main *first* move, and vice-versa, suggesting the model considers longer-term consequences of alternative moves. We note that the log odds reduction range is smaller than for the one branch case, in large part because the model's odds are spread between different branches.

Furthermore, our analysis of the L12H12 attention head in the context of alternative moves and checkmate scenarios (detailed in Appendices F and G) indicates that L12H12 strongly privileges moving information from the third to the first move square in the principal variation, even for puzzles where Leela does not choose this as the best move, as long as the puzzle set matches the pattern $(\cdots)$AAC mentioned above. In case of two branches with this pattern, head L12H12 appears to move information "backward in time" independently for each branch, without cross-attention behavior (see Figure 31). In scenarios where multiple checkmates are possible, L12H12 shows less clear attention patterns that span across different move branches, suggesting a sophisticated evaluation of multiple winning lines.

## 4. Related Work

Recent work in mechanistic interpretability has employed techniques like activation patching (Vig et al., 2020; Meng et al., 2023), probing (Hewitt and Liang, 2019; Gurnee et al., 2023), and ablation (McGrath et al., 2023) to understand model behavior. These approaches have been applied to game-playing models, including Othello (Li et al., 2023; Nanda et al., 2023) and chess (Karvonen, 2024), revealing how models represent and manipulate game states. Our work extends these techniques to understand look-ahead planning in chess, providing insights into how chess-playing transformers process information about future states.

Following AlphaZero's success (Silver et al., 2018), research has explored both the capabilities of chess networks (Czech et al., 2024; Ruoss et al., 2024) and their planning abilities (Jenner et al., 2024). This connects to broader work on planning in neural models (Men et al., 2024; Yao et al., 2024; Hao et al., 2023), where some studies have found evidence of multi-step planning (Chen et al., 2021), while others highlight potential limitations. Our controlled chess environment offers insights that may generalize to other domains requiring sophisticated planning capabilities.

## 5. Conclusion

In this study, we have explored the look-ahead behavior of the Leela chess model when solving chess puzzles, with a particular focus on understanding how the model processes and utilizes information about future moves.

First, we demonstrate that the model can process information about board states up to seven moves ahead, though this capability becomes progressively weaker for more distant moves. The model's look-ahead behavior is highly context-dependent, varying significantly based on the specific puzzle set and whether the sequence leads to checkmate.

Second, we find evidence that the model processes some future moves using similar concrete internal mechanisms, particularly through specialized attention heads like L12H12. These mechanisms appear to be pattern-sensitive rather than timing-specific, suggesting the model has learned some general strategies for processing look-ahead information rather than just heuristic rules.

Third, our analysis reveals that the model considers multiple move sequences simultaneously, with different attention heads specializing in processing different types of positions. For instance, L12H12 shows stronger responses in checkmate scenarios, while L12H17 is more active in non-checkmate positions. This specialization suggests the model has learned to handle different tactical situations using distinct mechanisms.

Our methodological approach, combining activation patching, probing, and ablation techniques, provides complementary insights into the model's behavior. While patching reveals causally necessary information, probing shows that the model encodes additional information (such as opponent moves) that may be used more subtly. This multi-faceted analysis approach could prove valuable for future studies of planning behavior in other domains.

These findings have broader implications for our understanding of how neural networks can develop sophisticated planning capabilities through training. The emergence of specialized components and general pattern-matching mechanisms, without explicit programming, suggests potential approaches for developing AI systems capable of strategic planning in other domains.

Future work could explore how these look-ahead capabilities generalize to other chess positions not present in the training data, or to modified versions of chess with slightly different rules. Additionally, investigating whether similar mechanisms emerge in neural networks trained on other strategic games or real-world planning tasks could provide valuable insights into the generality of these findings.

## Impact Statement

Understanding how models develop look-ahead capabilities and handle complex decision trees could inform the development of AI systems for other strategic tasks, and may help improve our understanding of how these capabilities may generalize, or fail to do so, in novel scenarios.

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

## A. Starting squares

To investigate whether the starting squares play a significant role in the model's behavior, we conducted a detailed analysis of the residual effects of patching these squares. We modified our puzzle set notation to account for the starting squares, where for a set $s_1 s_2 \cdots s_n$, the odd indices represent the squares the piece in play starts in, and the even indices represent the squares the piece moves to.

Our analysis focused on 3-move puzzles (n=6) to maintain consistency with previous studies and simplify the interpretation of results. Figure 7 presents the residual effects for the puzzle set 112, split into subsets based on the similarity between the starting squares.

The results demonstrate that the log odds reduction observed is not significantly different for any of the subsets when compared with the baseline results for puzzle set 112. This consistency across different starting square configurations suggests that the starting squares do not play a critical direct role in the model's decision-making process for these puzzles.

Based on these findings, we concluded that it was unnecessary to disentangle the effect of different starting square configurations when performing activation patching, probing, or ablation in subsequent analyses. This simplification allows us to focus on the more influential aspects of the model's behavior, particularly the squares to which pieces move during the course of play.

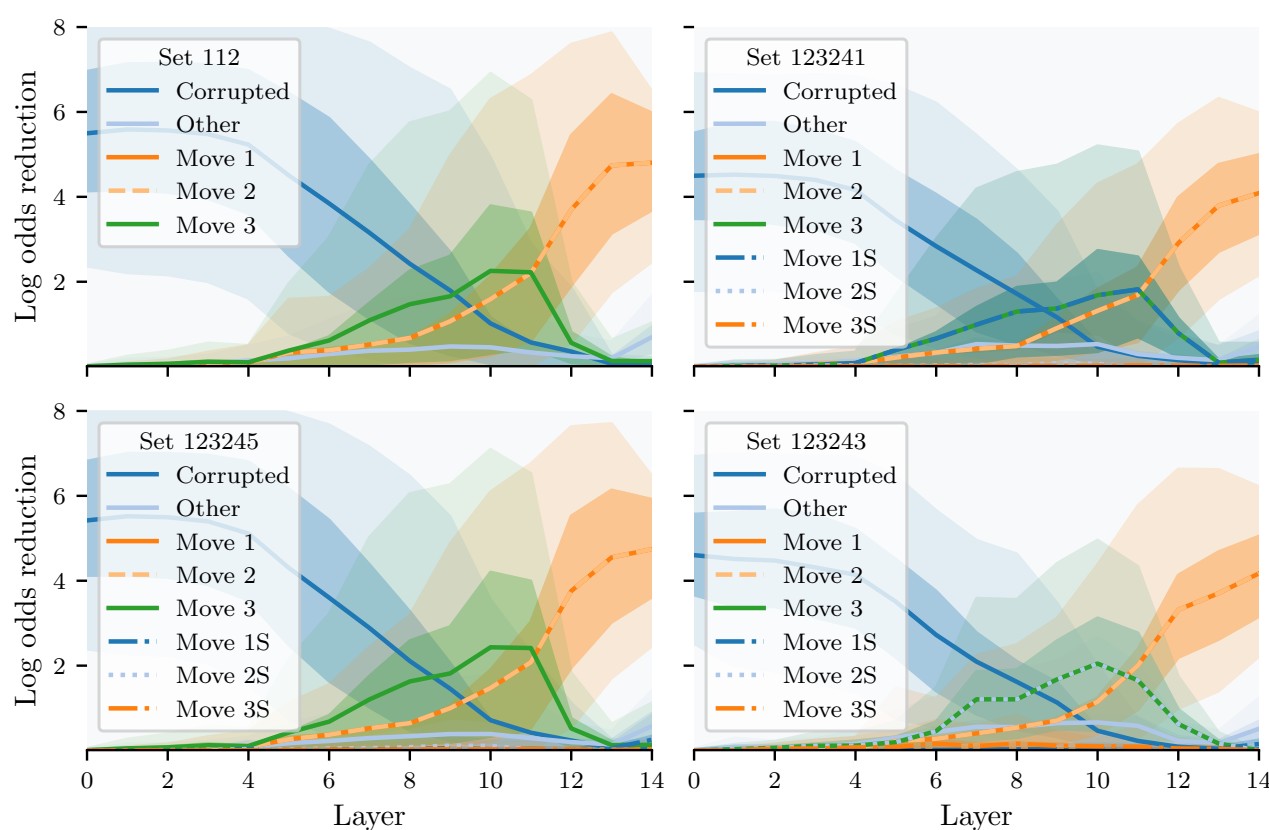

Figure 7: Residual effects of the starting squares for the puzzle set 112. The baseline analysis is replicated in the top left plot, while 3 different subsets of the original set are shown in the other plots. Using our puzzle set notation, we have the decoupling $\underline{112} \rightarrow \{123\underline{245}, 1\underline{23}\underline{241}, 1\underline{23}\underline{243}\}$ (the underlined digits denote the main move squares, with the other digits marking the starting squares). Not all possible sets are represented. Note that the effect of the starting square is not significant for any of the sets.

## B. Residual stream patching for 5-move and 7-move puzzles

In this section, we show the activation patching results for the remaining puzzle sets with 3, 5 and 7 moves. In Figure 8, the 112 and 123 set plots reproduce the results from Jenner et al. (2024), with the slight change that we do not include puzzles of higher move count. For example, the set 11234 is not counted as part of 112, as was the case in Jenner et al. (2024). Nonetheless, the puzzles with more moves have a lower count in the puzzle dataset, so the results are not too different from those previously observed.

In most patching plots in Figures 2 and 8 and Appendix B, there is a marked uptick in log odds reduction coming from the remaining squares for the last layer. A closer inspection reveals that this usually corresponds to the model's chosen first move square for the corrupted puzzle version used for patching.

The hypothesis presented in Section 3 is that the model's behavior is highly dependent on the puzzle set. We reiterate the hypothesis are, as follows:

*Hypothesis* 1. The effect of patching a move square is quantitatively different between certain puzzle sets. In increasing order of effect size, we note puzzle sets of the form $(\cdots)\text{ACC}(\cdots)$, $(\cdots)\text{ABC}(\cdots)$, and $(\cdots)\text{AAC}(\cdots)$.

In general, we exclude from the hypothesis puzzle sets where the odd move squares are not distinct.

As mentioned in Section 3, we hypothesize that the model may be using similar mechanisms as in the third move analysis. For the 3-move puzzle sets, we note a patching effect size, in increasing order, for 122, 123, and 112 (111 is excluded, as the first and third move squares are the same). In fact, we note the following orderings in patching effect size:

- Move 3: 122 < 123 < 112. Hypothesis holds. See Figure 8.

- Move 5: $(\cdots)\text{ACC} < (\cdots)\text{ABC} < (\cdots)\text{AAC}$ (in particular, (11)122 < (11)123 < (11)112, (12)344 < (12)345 < (12)334). Hypothesis holds. See Figures 9 and 11.

- Move 7: (1123)344 < (1123)345 ≃ (1123)334, (1123)455 < (1123)456 ≃ (1123)445. Hypothesis holds somewhat, conditional on the puzzle sets having the same prefix. See Figure 13.

Hypothesis 1 seems to hold somewhat for earlier moves when the puzzle set suffix is the same. We have:

- Move 3: 123(44) < 112(33), 123(45) < 112(34), 123(33) < 112(22), 123(4567) < 112(3456). The hypothesis does not strongly hold for the case 122(23) ≃ 123(34) ≃ 112(23).

- Move 5: (11)123(45) < (11)112(34), (11)233(34) < (11)234(45), (11)233(44) < (11)234(55), (11)233(45) ≃ (11)234(56).

Additional activation patching results are shown in Figures 10 and 12. Some of the theoretically possible puzzle sets are not shown because they are unlikely configurations. The puzzles shown have a sample size of at least 50 puzzles.

We showcase puzzle set 1123456 in Figure 14. We note that the model's log odds reduction as a result of patching the seventh move is non-negligible but relatively small, indicating that it is likely at the limit of the model's ability to look ahead. Moreover, the probing results in Figure 3 suggest that the model does contain information about the seventh move square, but the probe's performance is only slightly better than for the random chess model. Additional 7-move puzzle sets can be seen in Figure 13.

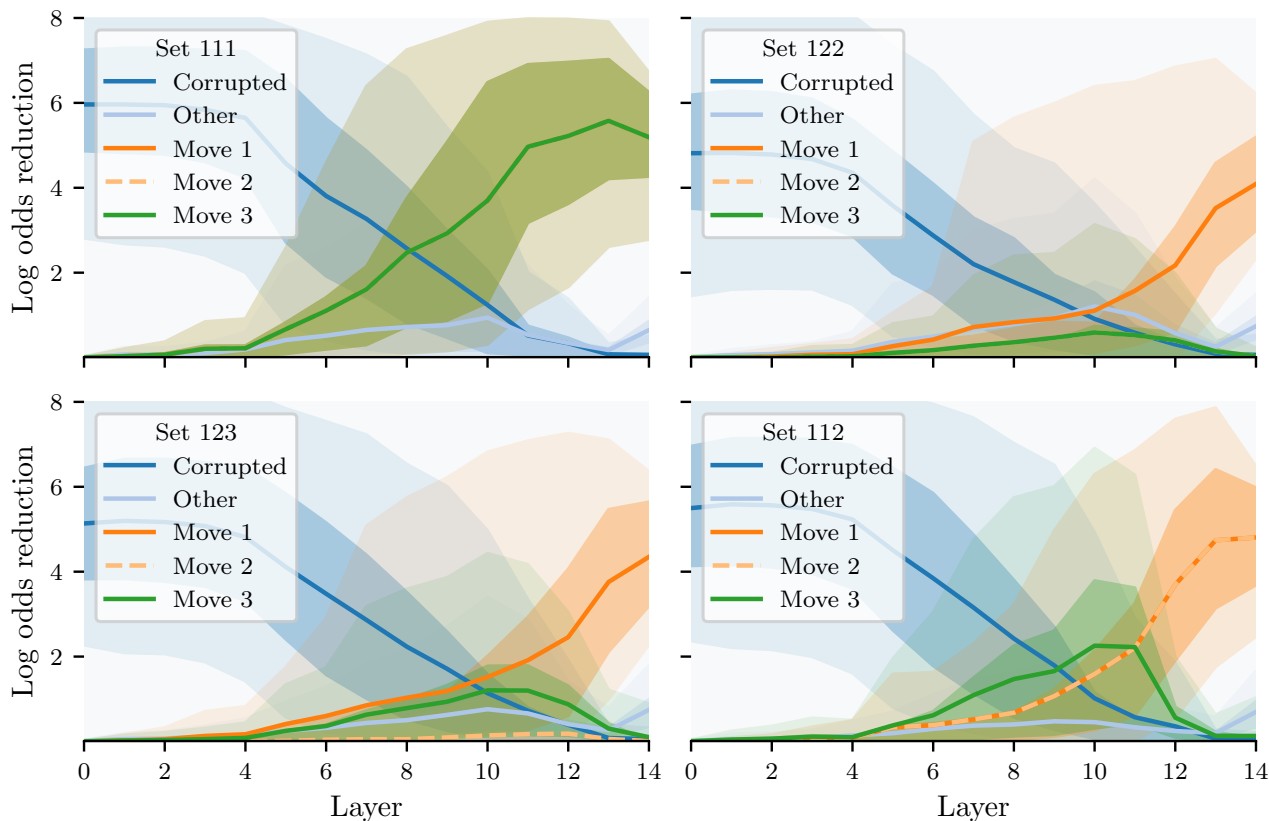

Figure 8: Log odds reduction of the correct move as a result of residual stream patching, for puzzles with 3 moves. "Corrupted" indicates the patched square from the corrupted board. The label $i$ indicates the move square for the $i$-th move. "Other" indicates the contributions of the remaining squares. Dashed lines indicate opponent moves. The 50% and 90% confidence intervals are displayed using darker and lighter colors, respectively.

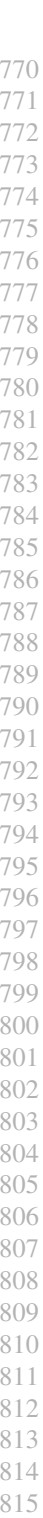
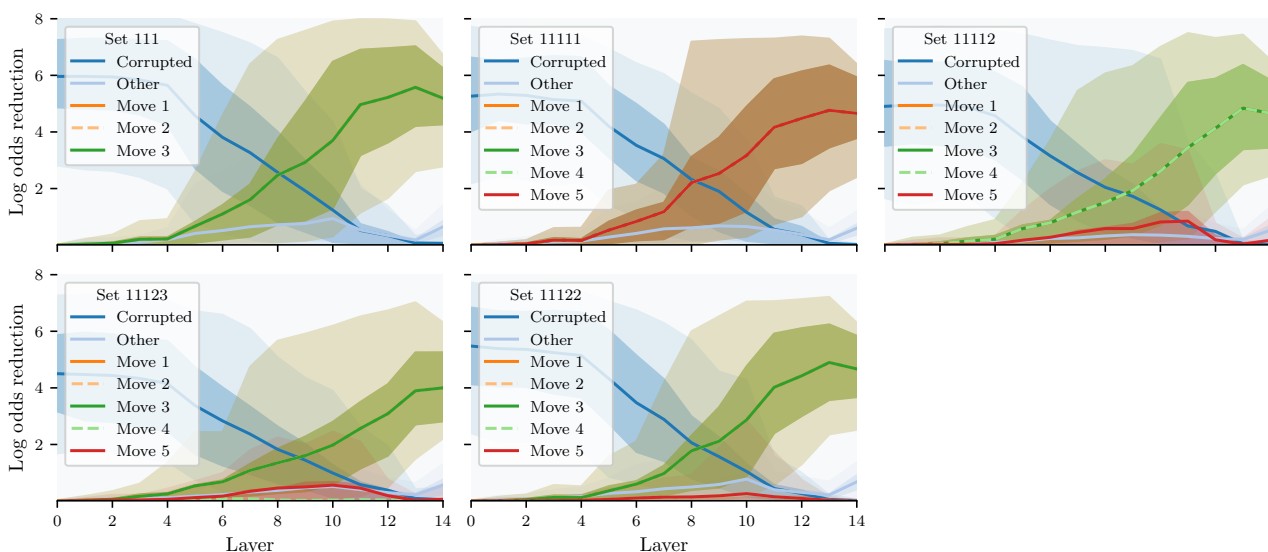

Figure 9: Log odds reduction as a result of residual stream patching, for puzzle sets of the form 111XY. The 3-move puzzle set 111 is shown on the top left, for comparison. As in Figure 2, we note that the impact of patching the fifth move square varies considerably from puzzle to puzzle, but is consistent with the hypothesis presented.

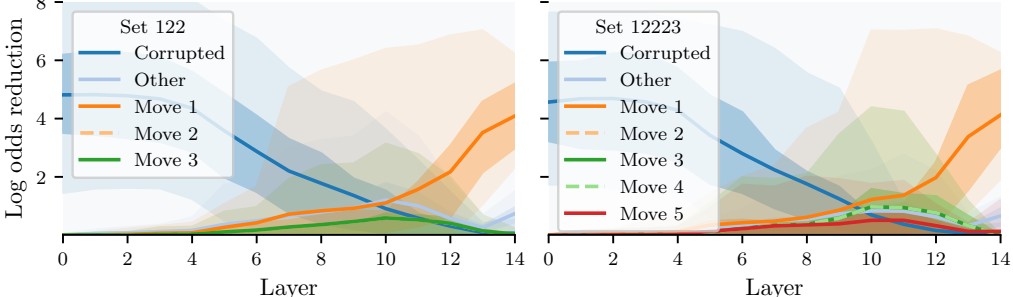

Figure 10: Residual effects the puzzle set 122 and 12223.

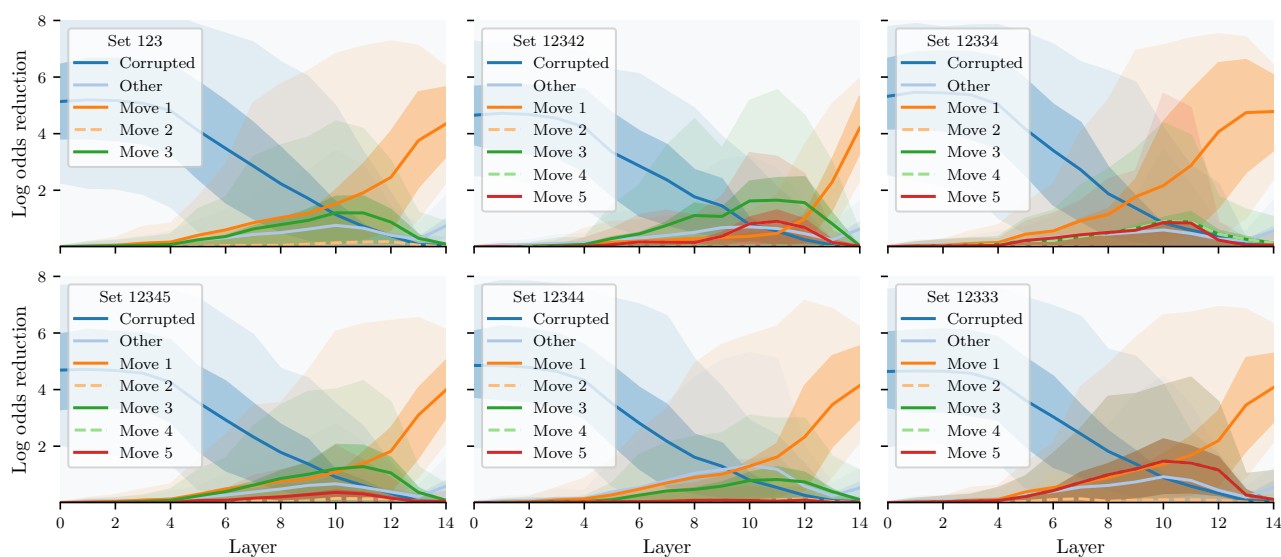

Figure 11: Residual effects of puzzle sets of the form 123XY. The 3-move puzzle set 123 is shown on the top left, for comparison. As in Figure 2, we note that the impact of patching the fifth move square varies considerably from puzzle to puzzle.

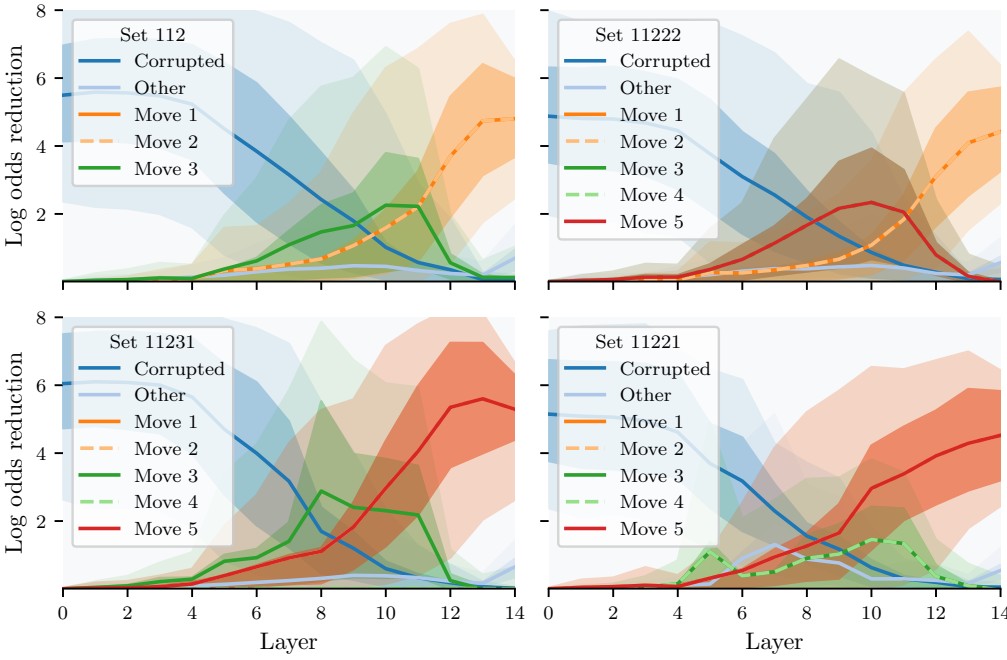

Figure 12: Residual effects for the remaining puzzle sets of the form 112XY, not shown in Figure 2. The 3-move puzzle set 112 is shown on the top left, for comparison. In these puzzle sets, it is not possible to distinguish the effect of patching the fifth move square directly, as it equals either the first or the third move square. Nonetheless, we note that puzzle sets 11231 and 11221 respond differently to patching the third move square. For sets where the 1st and 5th move square are the same, the effect of patching that square is more pronounced.

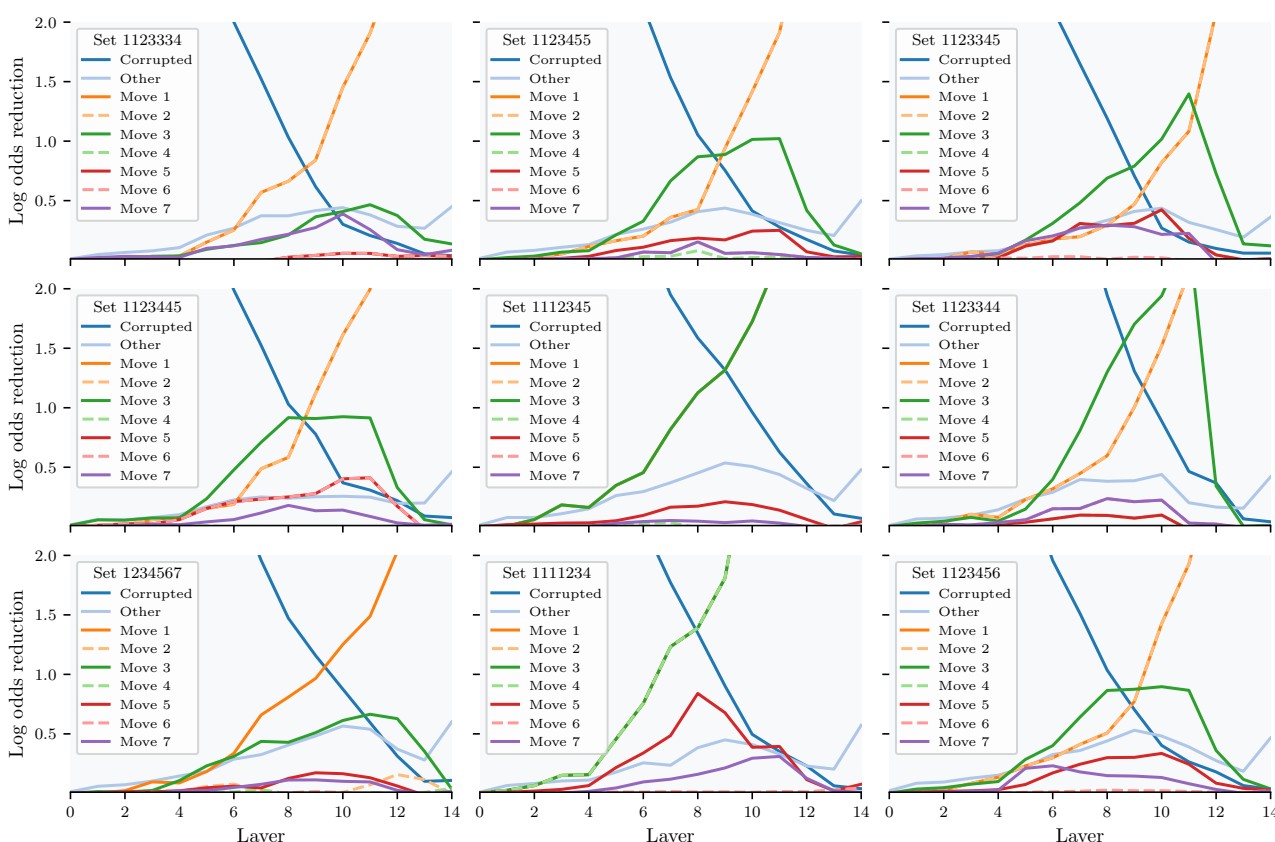

Figure 13: Residual effects for the puzzle sets with 7 moves. The effect of patching the seventh move square is small but not negligible for most of the puzzle sets, but its importance varies considerably.

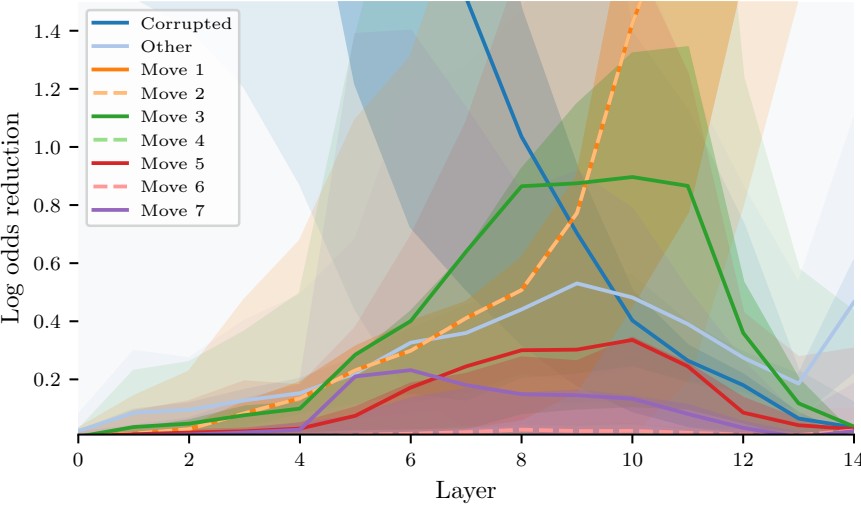

Figure 14: Log odds reduction of the correct move as a result of residual stream patching, for puzzles with 7 moves. "Corrupted" indicates the patched square from the corrupted board. The label $i$ indicates the move square for the $i$-th move. "Other" indicates the contributions of the remaining squares. The 50% and 90% confidence intervals are displayed using darker and lighter colors, respectively.

## C. Probing results

Our probing analysis provides additional insights into the model's ability to encode and utilize information about future moves. We conducted probing experiments on various puzzle sets to complement our activation patching results and gain a more comprehensive understanding of the model's internal representations.

Figure 3 presents the probing results for the puzzle set 1123456. The probe's accuracy shows a clear decreasing trend as the move square becomes increasingly distant from the present state. This decline in accuracy is particularly pronounced for the 7th move square, suggesting that while the model does encode some information about very distant future moves, this information becomes increasingly uncertain or difficult to extract.

Figure 15 shows the probing results for the puzzle set 12345, offering insights into how the model encodes information about both player and opponent moves. Several key observations can be made:

- The probe can find both player and opponent move squares with high accuracy, generally peaking at layer 13. This suggests that the model encodes information about opponent moves, even though activation patching does not show a strong direct response for these squares.

- The probe's accuracy decreases as the predicted move becomes more distant from the present state, consistent with our observations from activation patching.

- Interestingly, the 4th move (an opponent move) seems more difficult to predict than the player's 5th move. This could indicate that the model's representation of opponent moves is less direct or more uncertain than its representation of the player's own future moves.

- The probe's accuracy for the random chess model is notably higher for the first and second move squares, possibly reflecting some inherent biases or common patterns in chess openings.

These probing results complement our activation patching findings by revealing that the model does encode information about future moves, including opponent moves, even when this information does not have a strong direct effect on the model's output. This suggests that the model's internal representations are rich and multifaceted, capturing various aspects of potential future game states.

The discrepancy between probing and activation patching results, particularly for opponent moves, highlights the complexity of the model's decision-making process. It suggests that while information about opponent moves is present in the model's representations, it may be utilized in more subtle or indirect ways than information about the player's own moves.

These findings underscore the importance of using multiple analysis techniques to gain a comprehensive understanding of the model's internal workings and decision-making processes.

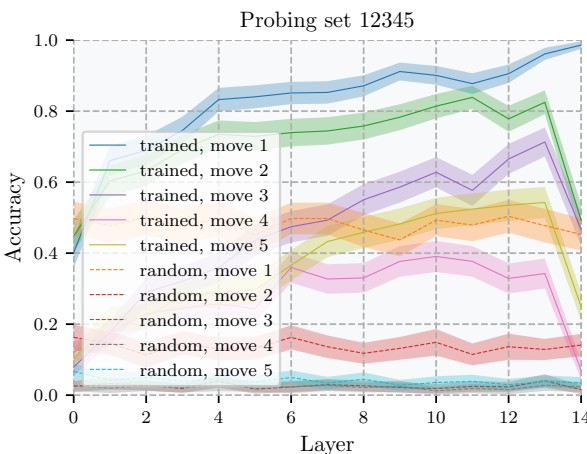

Figure 15: Probing the model for the puzzle set 12345. While activation patching does not seem to lead to a strong response for opponent move squares, the probe can find both the player and opponent move squares with high accuracy, generally peaking at layer 13. These results suggest that the model is encoding information about the opponent's moves in a less direct way than the player's moves. We observe the probe's accuracy decreases as the model becomes increasingly more distant from the present. A notable exception is move 4, which seems to be harder to predict than the player's fifth move. The probe's accuracy for the random chess model is notably higher for the first and second move squares.

# D. Ablation results

This section presents a detailed analysis of the ablation results for various attention heads, with a particular focus on L12H12, which appears to play a crucial role in the model's look-ahead behavior.

The L12H12 head behavior (Figures 16 to 18) is consistent with the following observations:

- The head moves information from the 3rd to the 1st move square for set 112, and for all puzzle sets 112XY and 112VWXY, and to a lesser extent for the sets 123XY. The weakest case is the set 11223, as expected, since the 5th move behavior takes precedence.

- The head moves information from the 5th directly to the 1st move square for the puzzle set 11223 and 11112 (pattern $(\cdots)$AAC), and to a lesser extent for the set 11234 (pattern $(\cdots)$ABC). For 7-move puzzles, the effect is strongest for sets 11112XY, and to a lesser extent for 11234XY.

- The head moves information from the 5th to the 3rd move square for the puzzle sets 11223 and 12223 (pattern $(\cdots)$AAC).

- The head moves information from the 7th to the 1st move square for the puzzle sets 1123334, 1111234, and 1123456 (patterns $(\cdots)$AAC and $(\cdots)$ABC).

- The head does not directly move information from the 3rd move square for puzzle sets 122 or 122XY. It also does not move information from the 5th move square for puzzle sets $(\cdots)$ACC (such as 11222, 12344).

Nonetheless, the hypothesis is not compatible with the set 12334, since we would expect to observe behavior in between sets 12223 and 11223, and to mainly move information from the 5th to the 3rd or 1st move square, instead of from the 3rd to the 1st move square. We would also expect some effect from the 3rd to 1st move square for 12344. For 7-move puzzles, we observe no 3rd to 1st effect for 123VWXY sets.

Overall, head L12H12 appears to satisfy the hypothesis presented in Section 3, and noted for the residual stream patching analysis, where we observed that the model tends to move information from future move squares to earlier move squares. Specifically, this head seems to prioritize moving information from the 3rd to the 1st move square for patterns like AAB and ABC, from the 5th to the 1st or 3rd move square for patterns like AAC, and from the 7th to the 1st move square for patterns like AAC and ABC.

For other attention heads:

- L12H17 (Figure 19) appears to move information "backward in time" for puzzle sets of the form AABCD, where C is different from D, and D is preferably equal to A. In sets of the form AABCA, the model relies more heavily on L12H17 than on L12H12.

- L13H3 (Figure 20) seems to move information "backward in time" for puzzle sets of the form AABCD, where either C=D or B=C.

- The roles of L11H10 and L11H13 (Figure 21) are less clear based on the ablation results alone.

Our detailed analysis of these attention heads reveals several important insights into how the model processes future move information. First, we find that certain heads specialize in moving information from future move squares to the first move square, responding to specific patterns that are round-insensitive - that is, the same pattern may apply to moves 1-2-3, moves 3-4-5, or moves 5-6-7. This suggests the model has learned general pattern-matching mechanisms rather than position-specific rules.

Different attention heads appear to specialize in different types of common patterns. For instance, L12H12 is particularly active in checkmate scenarios, while L12H17 shows stronger responses in non-checkmate positions. This specialization indicates that the model has learned to process different types of tactical situations using distinct mechanisms.

These findings have broader implications beyond this specific chess model. They demonstrate how a neural network can learn to develop specialized components for processing look-ahead information through training, without explicit programming of such capabilities. The emergence of these general pattern-matching mechanisms suggests the model may be able to handle novel positions not seen during training. Additionally, this analysis provides a case study of how detailed attention head analysis can reveal the development of sophisticated information processing strategies in trained models, insights that may extend to models in other strategic planning domains.

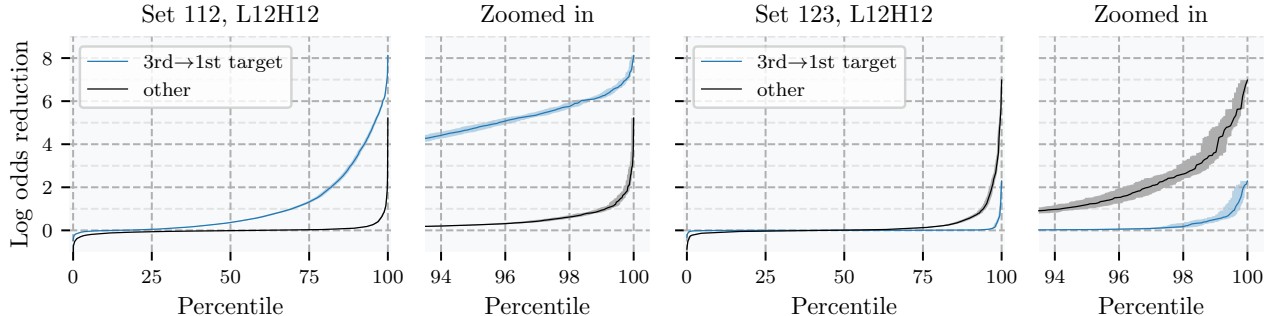

Figure 16: Ablation results of the L12H12 head for the 112 and 123 move analysis.

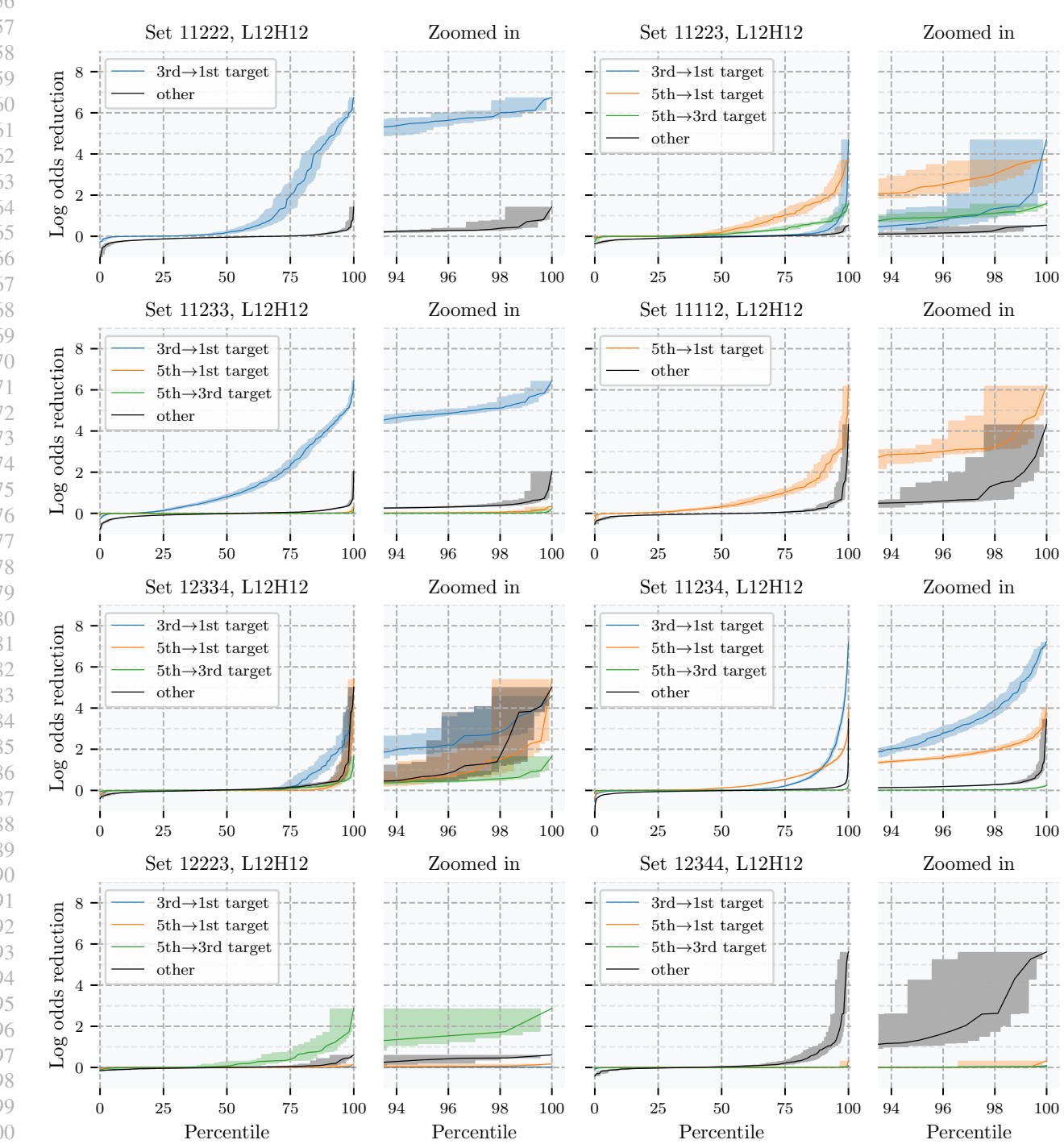

Figure 17: Ablation results of the L12H12 head for sets with 5 moves. This head's role varies significantly between the sets. For the sets 11223 and 11234, the head plays a significant role in moving information from the fifth to the first move square. For the sets 11223 and 12223, it also moves information from the fifth to the third move square. For the sets 11222, 11233, and 12334, it mostly plays the known role of moving information from the third to the first move square. For the set 12344, it seems to be doing something else entirely.

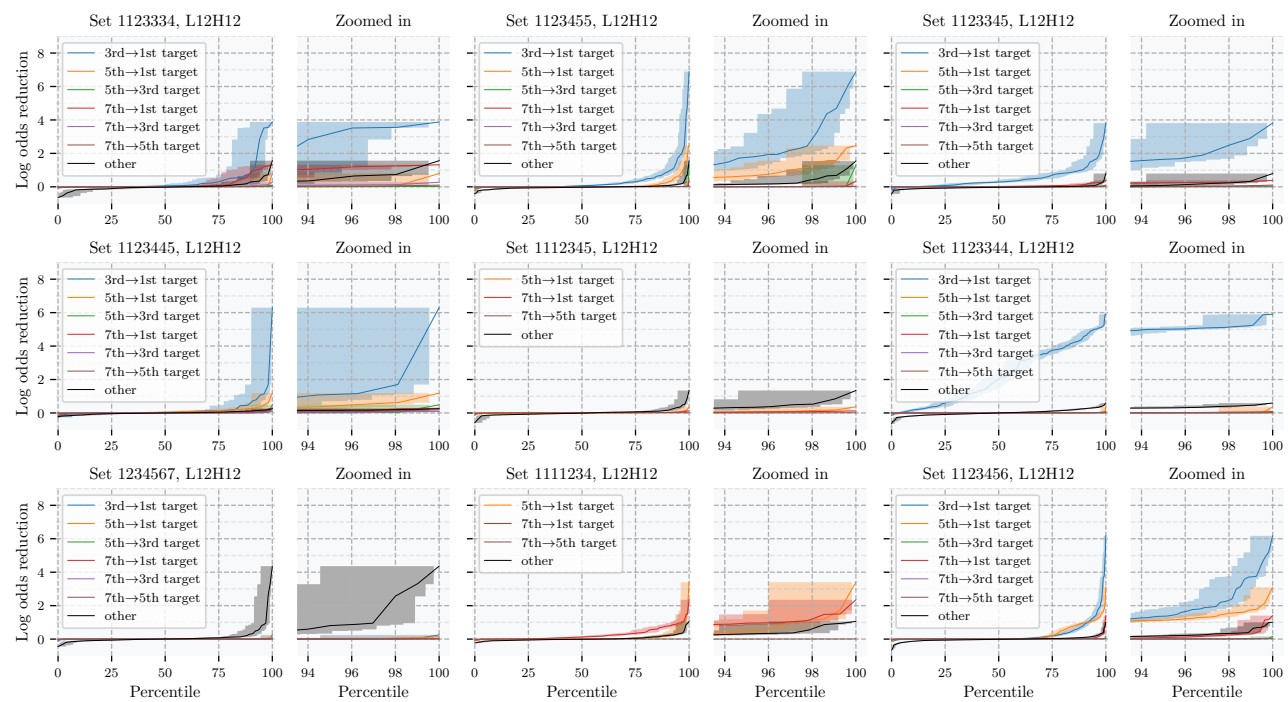

Figure 18: Ablation results for some puzzle sets with 7 moves, for head L12H12. The results are quite varied. For puzzle sets 1123334, 1111234, and 1123456, the head has a small but non-negligible role in moving information from the seventh to the first move square. For the most of the remaining puzzles, it mainly seems to move information from the fifth and third move squares to the first move square. For puzzle sets 1112345 and 1234567, the head appears to move information from and to unknown squares.

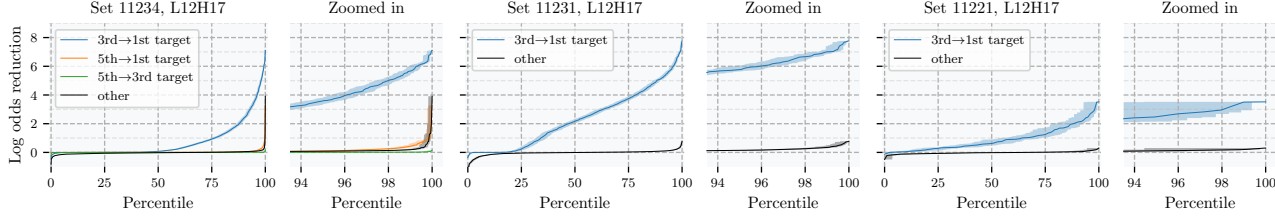

Figure 19: Ablation results for head L12H17, for the puzzle sets with 5 moves that seem to respond more strongly to the head being patched (see Figure 24). In all cases, the head plays a significant and almost exclusive role in moving information from the third to the first move square.

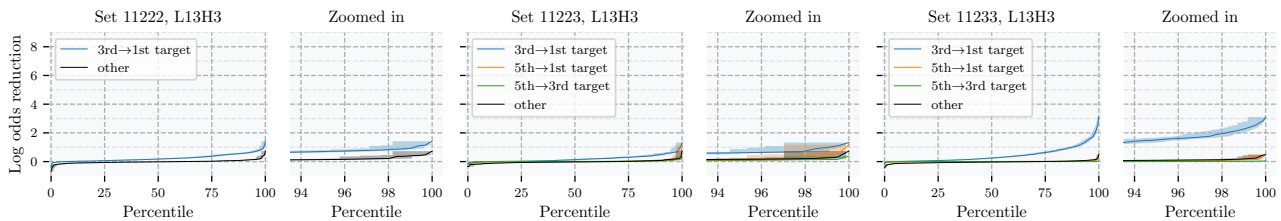

Figure 20: Ablation results for head L13H3, for the puzzle sets with 5 moves that seem to respond more strongly to the head being patched (see Figure 24). When compared to heads L12H12 and L12H17, the head L13H3 plays a less significant role in moving information from the third to the first move square.

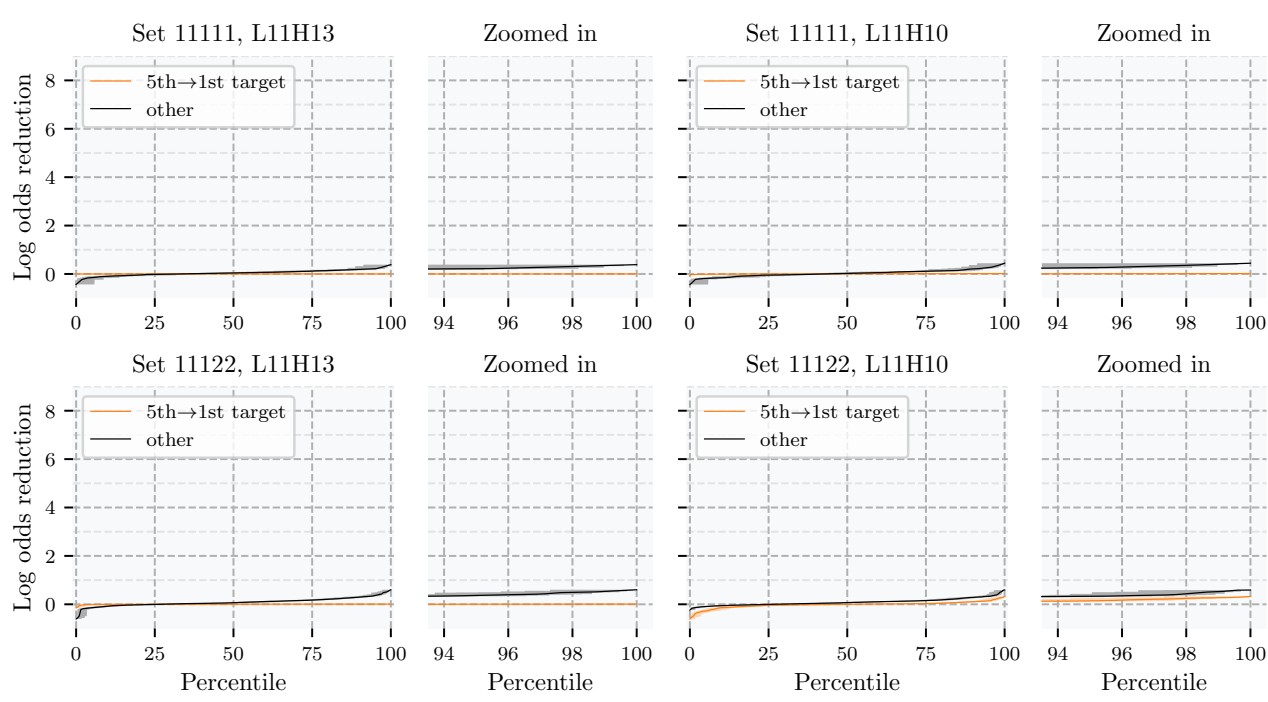

Figure 21: Ablation results for heads L11H10 and L11H13, for the puzzle sets with 5 moves that seem to respond more strongly to the heads being patched (see Figure 24). Surprisingly, the heads L11H10 and L11H13 seem to play a very minor role in moving information from and to squares of interest.

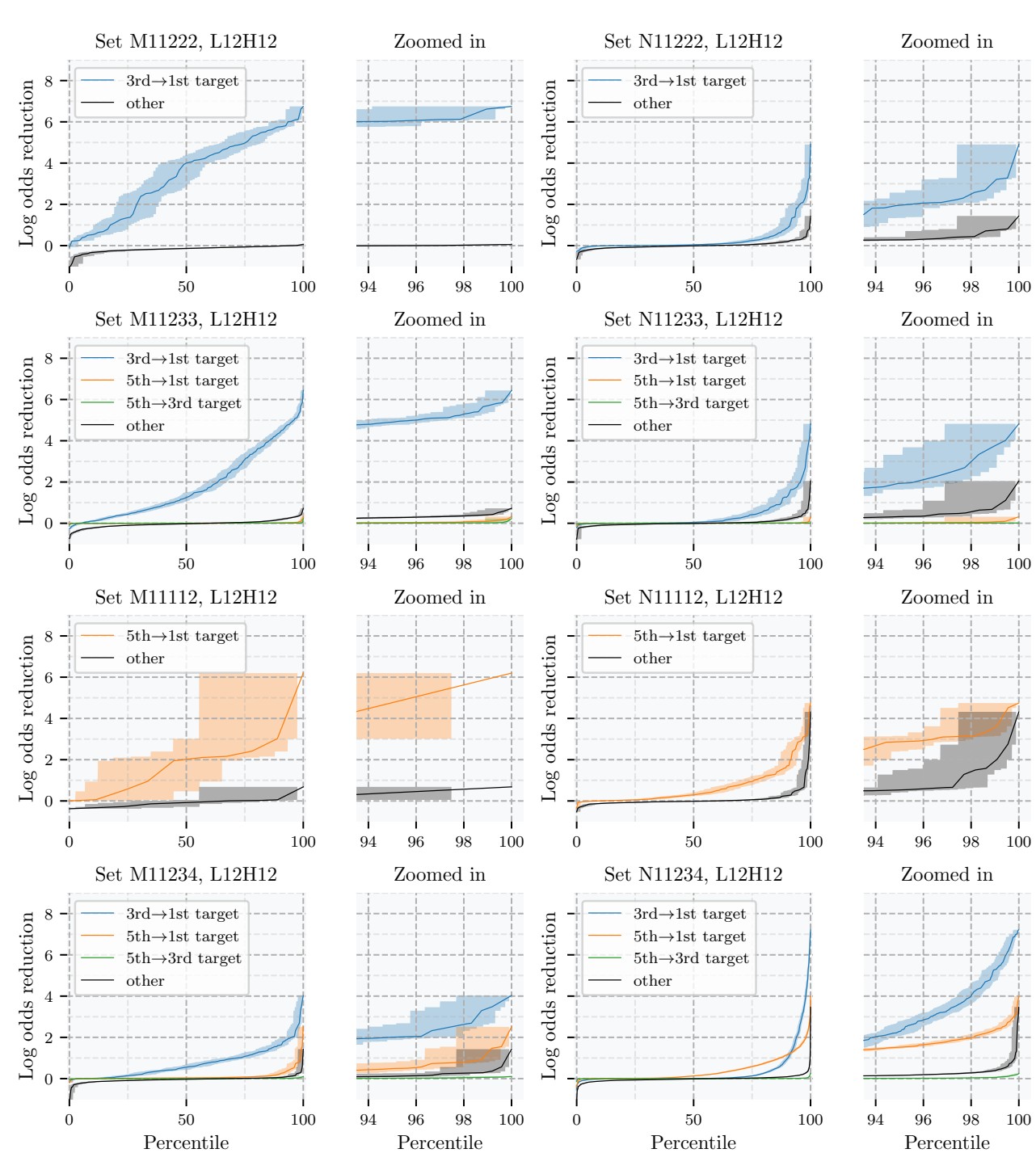

Figure 22: Ablation results for L12H12, for the puzzle sets with 5 moves that seem to respond more strongly to the heads being patched (see Figure 24). When decomposing by checkmate vs non-checkmate scenarios, we can observe that L12H12 plays a more significant role in moving information backward in time in the checkmate scenarios.

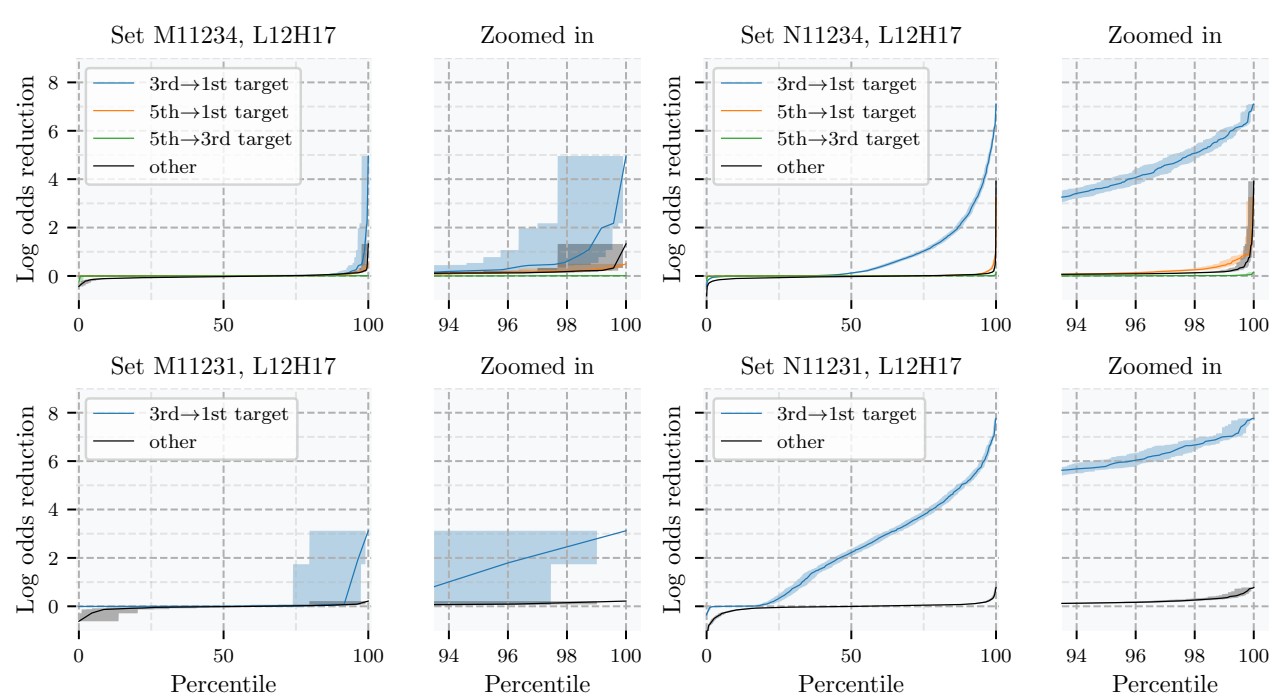

Figure 23: Ablation results for L12H17, for the puzzle sets with 5 moves that seem to respond more strongly to the heads being patched (see Figure 24). When decomposing by checkmate vs non-checkmate scenarios, we can observe that L12H17 plays a more significant role in moving information backward in time in the non-checkmate scenarios.

# E. Attention head patching

This section presents a comprehensive analysis of attention head patching results for various puzzle sets, providing insights into how different attention heads contribute to the model's decision-making process.

For 5-move puzzles (Figures Figure 24 and Figure 25), we observe distinct patterns:

- Strong response to L12H12 and L13H3: Some puzzle sets (e.g., 11223, 11233, 11234) show a strong response to patching these heads, suggesting their crucial role in processing these positions.

- Mixed responses: Puzzle set 11234 also responds strongly to L12H17, indicating a more complex interaction of attention heads for this set.

- Strong response to L12H17: Some sets (e.g., 11222, 12223) respond strongly to L12H17 patching but hardly to L12H12, suggesting different mechanisms at play for these positions.

- Weak or inconsistent responses: Some puzzle sets (e.g., 12233, 12234) do not show strong responses to any particular attention head, which may indicate more distributed processing or the involvement of other model components.

- Response to L11H10 and L11H13: Some sets (e.g., 11111, 11112) show responses to these heads, but ablation results suggest their role may be more subtle or indirect.

For 7-move puzzles (Figure 26), the patterns become more complex, potentially reflecting the increased difficulty in processing longer move sequences.

The analysis of checkmate vs. non-checkmate scenarios (Figures Figure 27 and Figure 28) reveals significant differences in attention head responses between these two types of positions. This suggests that the model may employ distinct processing strategies for checkmate and non-checkmate positions, potentially reflecting the different strategic considerations involved in each case.

These results highlight the context-dependent nature of the model's attention mechanisms and the complex interplay between different attention heads in processing chess positions. They also underscore the importance of considering factors like move sequence length and the presence of checkmate possibilities when analyzing the model's behavior.

## E.1. Checkmate and non-checkmate scenarios

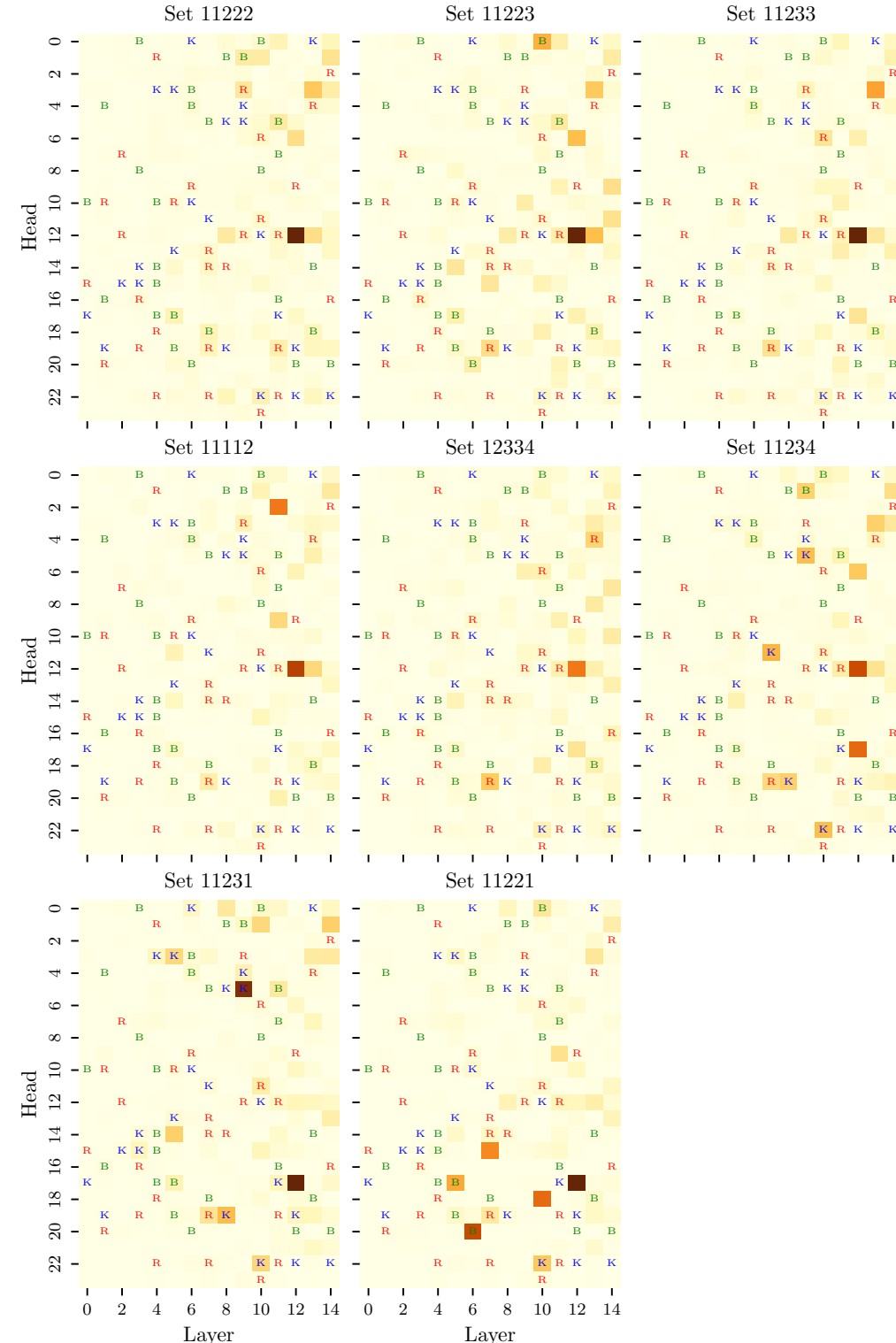

Figure 24: Attention head patching for some puzzle sets with 5 moves. In the top row, the model responds strongly to patching L12H12, and to a lesser extent L13H3 (see Figures 17 and 20). In the middle row, the response is more mixed, with puzzle set 11234 also responding to L12H17. In the bottom row, the model responds strongly to patching L12H17, and hardly responds to L12H12 (see Figures 19 and 20).

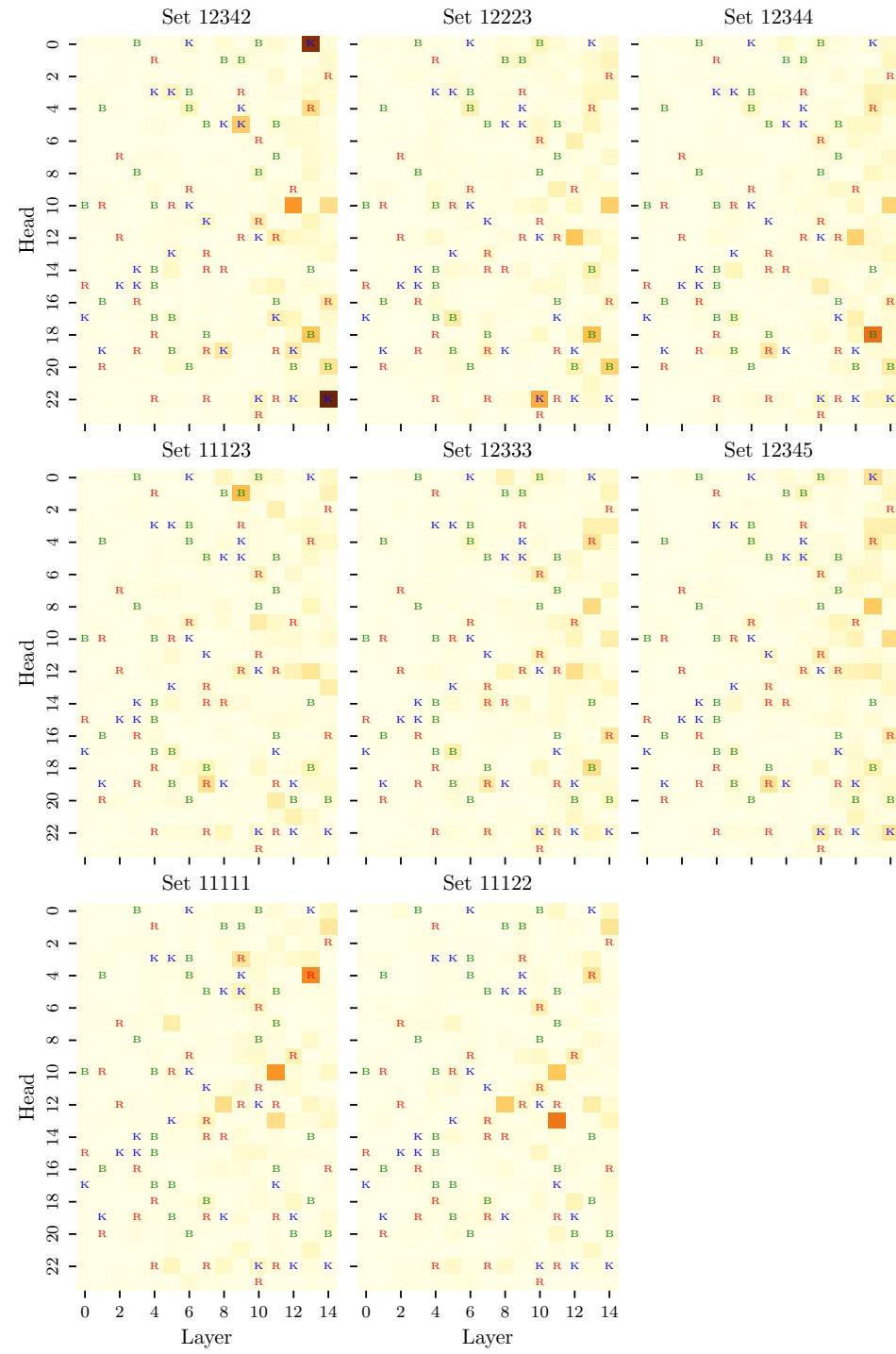

Figure 25: Attention head patching for the remaining puzzle sets with 5 moves. In the top row, some attention heads appear to strongly affect the model's behavior, but these do not appear to play a significant in other sets. The puzzle sets in the middle row do not seem to respond strongly to any particular attention head. In the bottow row, the model appears to respond somewhat to patching of heads L11H10 and L11H13. However, judging by the ablation results in Figure 21, these heads do not seem to play a significant role in the model's behavior.

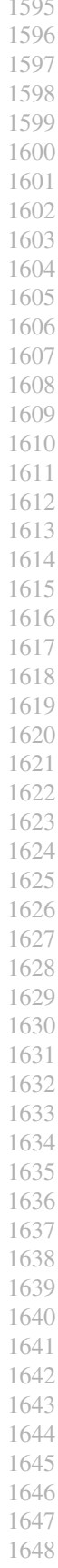

Figure 26: Attention head patching for some puzzle sets with 7 moves.

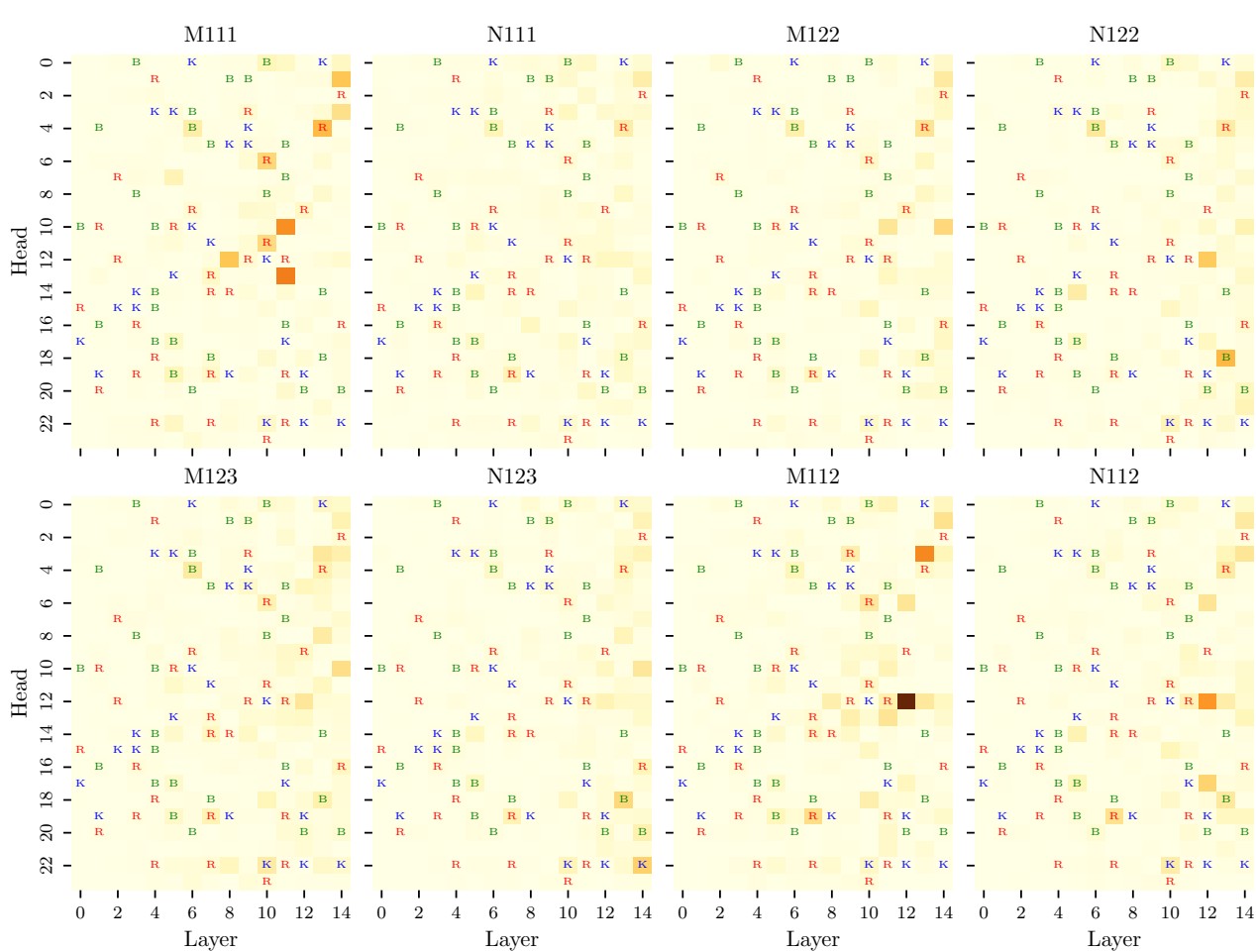

Figure 27: Attention head patching for puzzle sets with 3 moves, for both checkmate and non-checkmate scenarios. We can observe that patching the attention heads leads to notably different outcomes in each scenario.

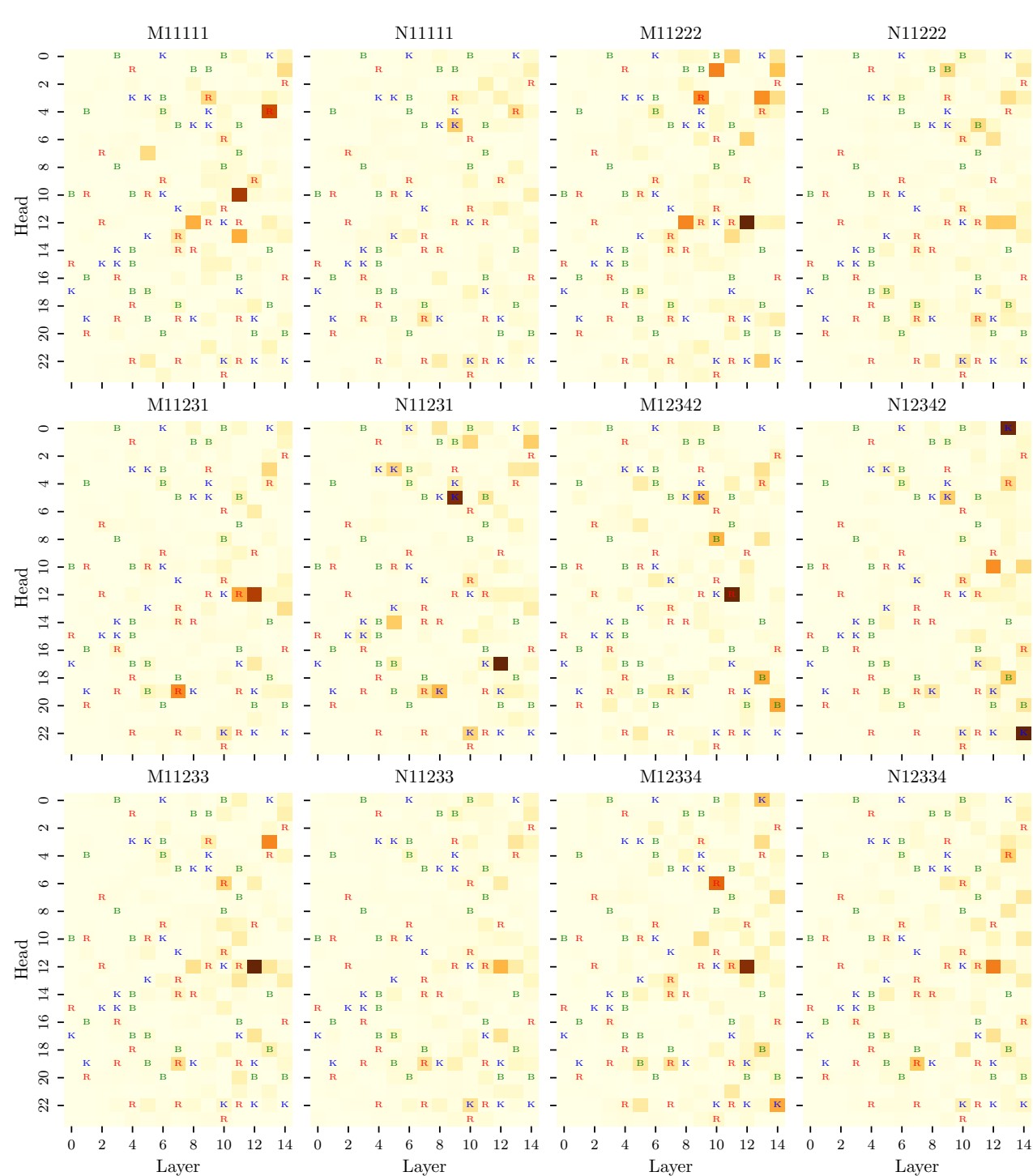

Figure 28: Attention head patching for some puzzle sets with 5 moves, for both checkmate and non-checkmate scenarios.

## F. Alternative move setup and additional results

This section details our approach to analyzing how the model considers alternative moves, focusing on puzzles with two distinct branches of play.

In order to study the alternative move analysis, we have to find puzzles that satisfy a significant number of constraints:

- **The puzzle's principal variation (PV) must have length 3.** This is in order to simplify the analysis, and remove higher future move squares from consideration.

- **The puzzle's PV must not be a checkmate.** In practice, we observe that not only does the model treat different puzzle sets differently, but it also seems to have a different behavior when a checkmate in 2 is a likely option, even if not the most likely.

- **The puzzle must have two distinct branches:**

  - **The model must be ambivalent between two first moves.** Each move should have a probability of around $1/2$. In practice, we impose a lower bound of $p = 0.3$ for the two moves.
  - **Given a first move, the model must be confident in the second move.** In practice, we impose a lower bound of $p = 0.7$ for the second move.
  - **Given the first two moves, the model must be confident in the third move.** In practice, we impose a lower bound of $p = 0.7$ for the third move.
  - **One of the branches must correspond to the PV.** Otherwise, the model cannot be said to be close to solving the puzzle, and it would be unclear to what extent the model's attention is due to the alternative move setup.

  The latter two conditions are mainly to ensure that the model's attention is not too spread out over relatively unlikely future moves. Essentially, we are interested in puzzles like the bottom example in Figure 29 (but without the checkmate scenario).

- **The two first and third move squares must all be distinct.** Otherwise, it would be impossible to distinguish the effects of the 4 squares in the analysis.

- **The puzzles should still be hard for the weaker model to solve.** The hardness threshold is maintained at 0.05, as in Jenner et al. (2024).

- **The weaker model should be confident in the second move.** The forcing threshold is maintained at 0.7, as in Jenner et al. (2024).

- **The corrupted puzzle versions should be viable for both branches.** Previously, we found the corrupted puzzles using only constraints with the PV moves. Here, we also require that the corrupted puzzles are viable for both branches. Otherwise, the corrupted puzzle may treat the branches differently, and lead to unclear results.

These constraints are highlighted in Figure 29. Regrettably, starting with the whole Lichess' puzzle dataset, these constraints reduce the original 4062423 puzzles to around 600 puzzles. In practice, we observe that about half to two thirds of the puzzles have differences between the probabilities assigned to the two branches' first moves that are non-negligible, and that may explain some of the limited log odds reductions observed in Figures 30 and 31.

These results, while based on a limited sample size due to our strict criteria, provide evidence that the model does consider alternative moves in its decision-making process. The varying effects across different puzzle sets suggest that this consideration is context-dependent.

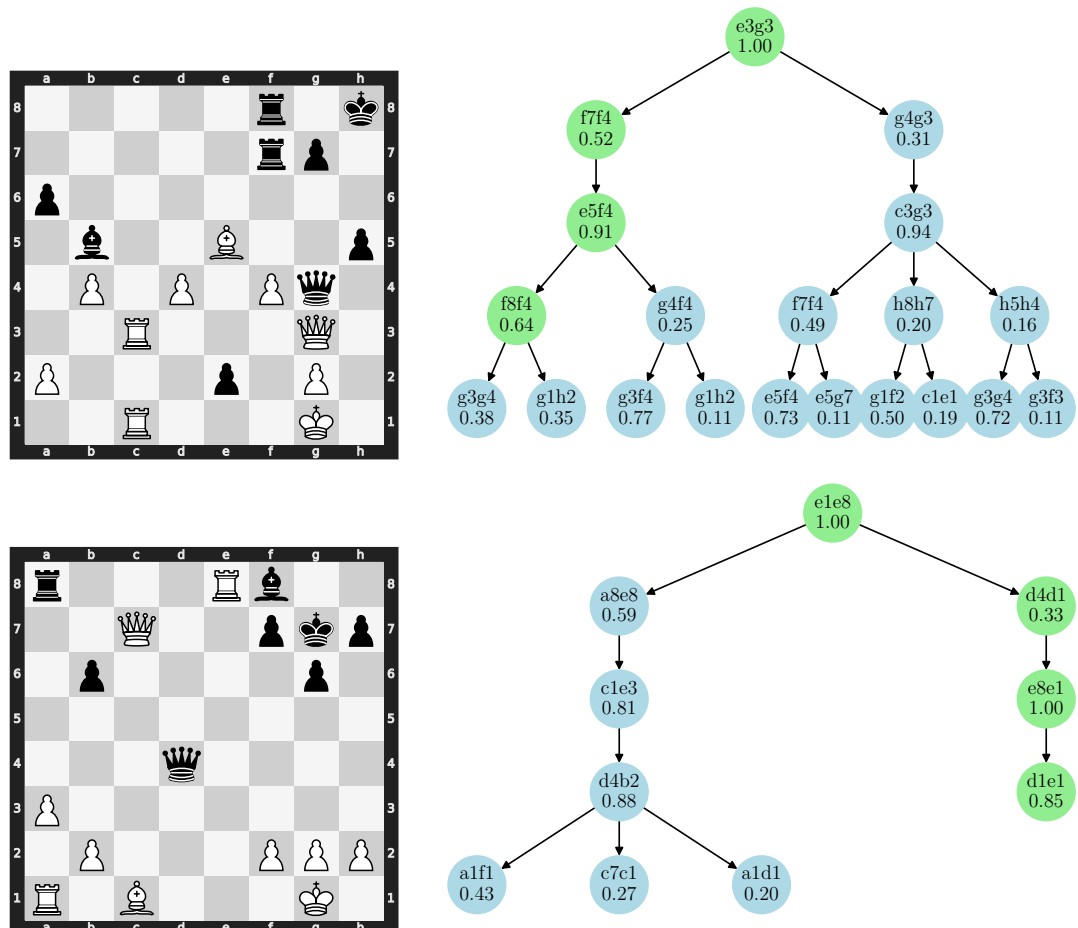

Figure 29: Instead of considering an arbitrary puzzle (top), where the number of branching moves can become very large, we focus on puzzles with two distinct branches. The boards correspond to the starting state of the two puzzles, after the zeroth move (top of game tree) is played. The green nodes mark the principal variation. Note that, for the bottom example, the Leela model does not choose the best move, but instead chooses an alternative move.

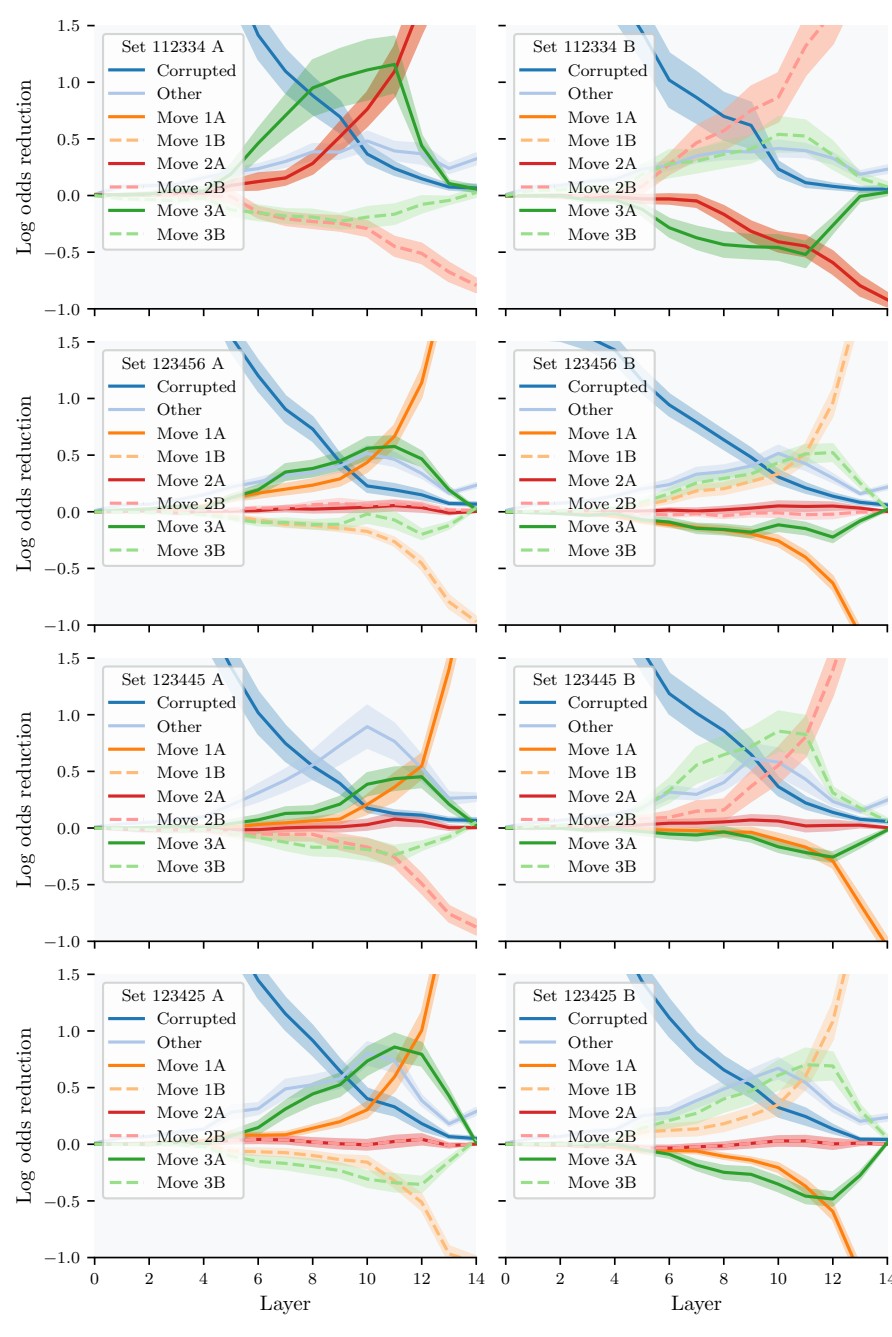

Figure 30: Patching results of the alternative move analysis. The log odds reduction for the next move for branch A (left) and branch B (right) are shown. Negative log odds reduction for branch A (resp. B) implies that patching the square improves the model's odds of choosing the main (resp. alternative) move branch. In this setup, the puzzle set label's first half denotes the most likely branch, and the second half denotes the second most likely branch. The number of examples is highly constrained, due to all the constraints imposed (see Appendix F for details). The model seems to consider alternative moves as one might expect. The effect of patching the alternative move squares (1B, 3B) seems especially pronounced for the puzzle sets for which L12H12 responds strongly. The bottom row is also reproduced in Figure 6.

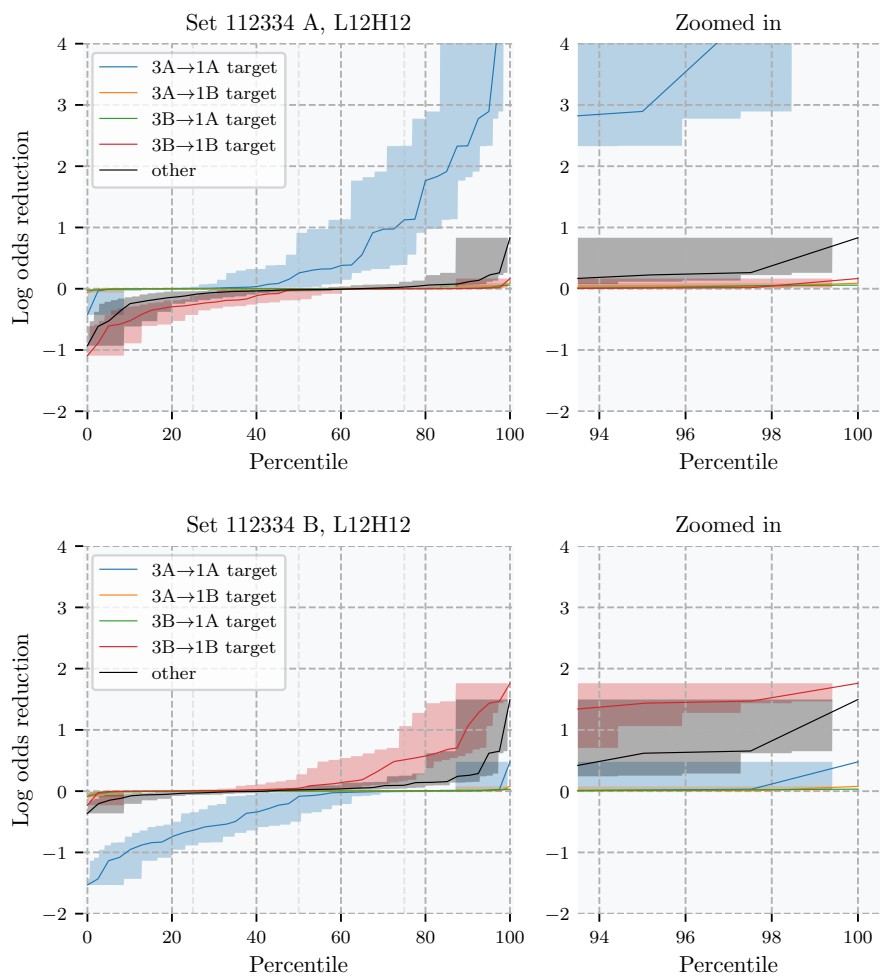

Figure 31: Ablation results for the alternative move analysis. The log odds reductions are shown for the next move when comparing against the branch A (top) and branch B (bottom) clean log odds. Negative log odds reduction for branch A (resp. B) implies that patching the square improves the model's odds of choosing the main (resp. alternative) move branch. We note that the head L12H12 appears to continue to focus on moving information from the third to the first move square, even when considering alternative moves. There does not seem to be significant cross-attention between the two branches, with both branches being processed independently.

## G. L12H12 and checkmate

While studying the L12H12 head in the alternative move analysis, we noted that the head seems to strongly privilege moving information from the third to the first move square in the principal variation, even for puzzles where the Leela chess model does not choose the principal variation as the best move. The main result can be seen in Figure 5, but here we specifically analyze this attention head in the alternative move setup.

### G.1. Different first moves

Upon further inspection, we noted that L12H12 seems to further prioritize scenarios involving checkmate. As a result, in the situation where the principal variation resulted in checkmate, but this was not the model's top move, L12H12 still mainly attended to the principal variation squares.

To further investigate this phenomenon, we looked at puzzles of the set 112 where both the first and second top moves resulted in checkmate in 2. Unfortunately, none of Lichess' 4 million puzzles seem to contain such puzzles. As a result, we produced a series of handcrafted puzzles, and studied the attention of the L12H12 head to each of the squares in both branches. Since this scenario is not present in the Lichess dataset, it is possible that this scenario is extremely unlikely, and that the Leela model has not encountered such scenarios during training. In fact, even for relatively simple handcrafted puzzles, the model does not always choose the checkmate in 2.

See Figures 32 and 33 for results. Interestingly, L12H12 not only shows the attention pattern g8→f7 and c8→d7 (corresponding to 3rd→1st, as expected), but there is also cross-attention between the 3rd move square and the 1st move square of different branches.

### G.2. Different third moves

We may also look at scenarios where puzzles (of the set 112) have two different possible third moves for a checkmate in 2, while having the same first and second moves. In the dataset `interesting_puzzles_all.pkl`, we find 10 puzzles of this type. The puzzle with the highest attribution values is shown in Figure 34.

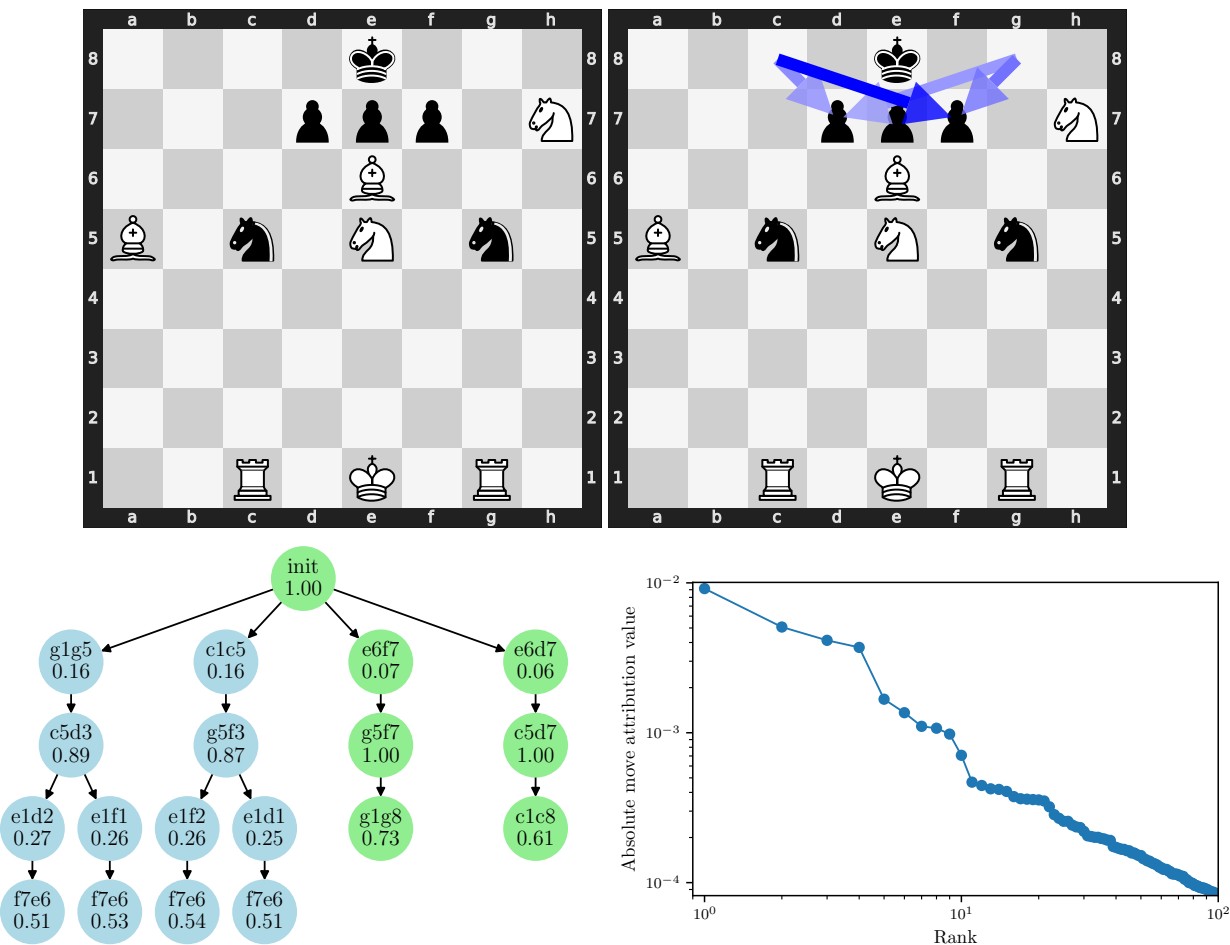

Figure 32: Simple puzzle where Leela fails to pick any of the checkmates in 2 moves. Nonetheless, the future move squares of those two game branches are still the main squares that L12H12 attends to, although weakly.

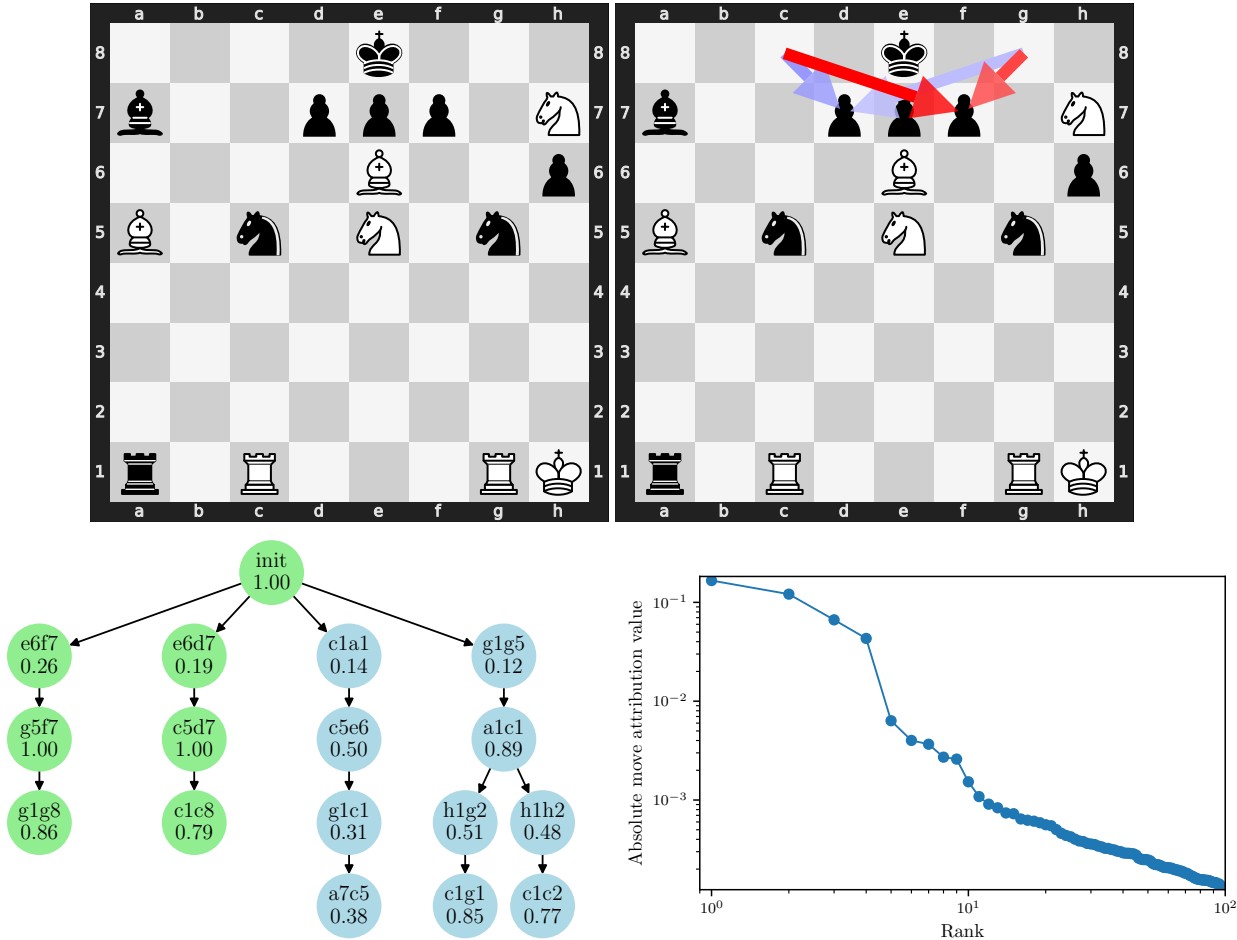

Figure 33: Modification of the puzzle in Figure 32 where the rook moves are discouraged, leading Leela to prefer the checkmate in 2 options. L12H12 attends to the relevant squares much more strongly in this case. Note the scale difference in the attribution plot when compared to Figure 32.

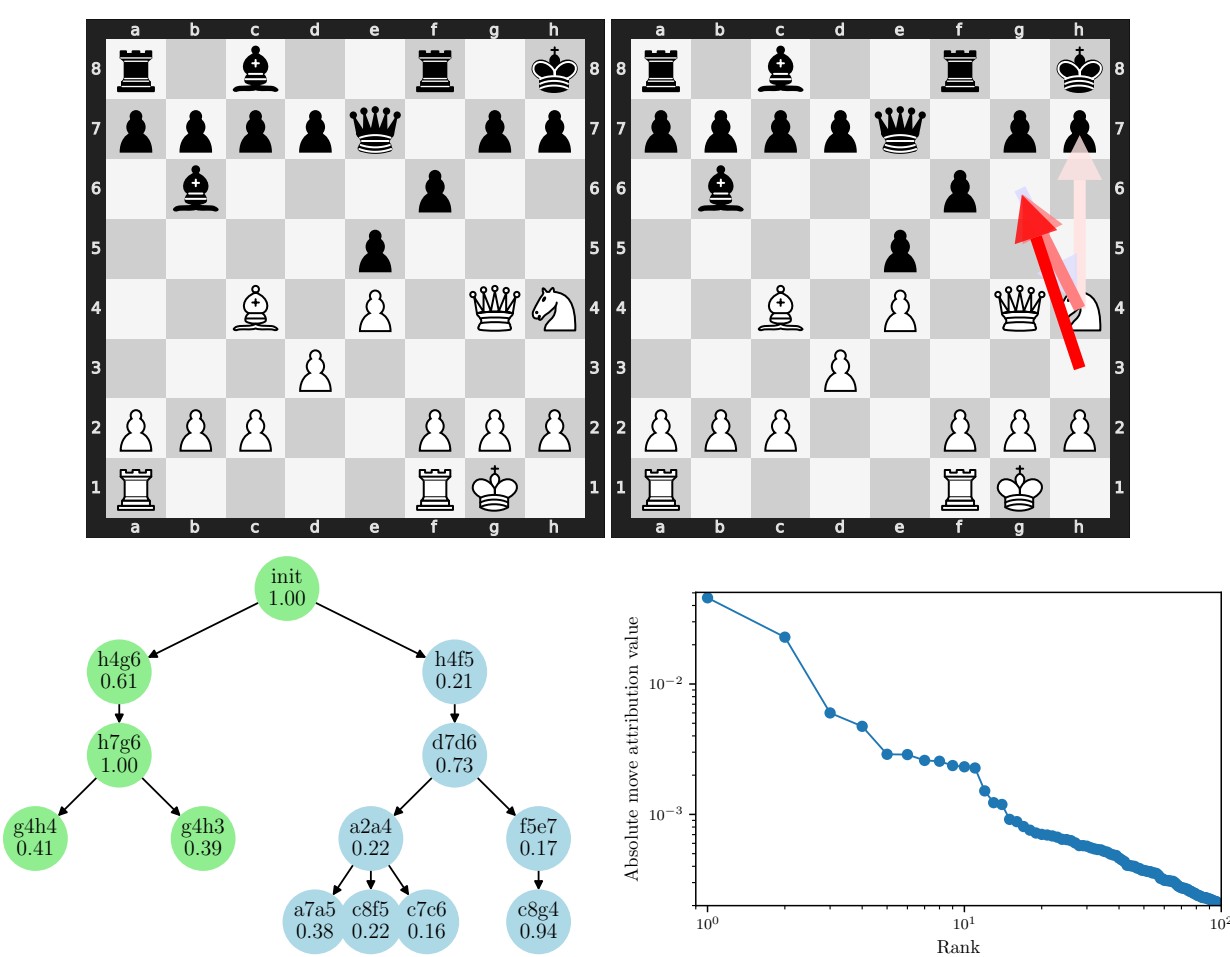

Figure 34: Puzzle of the set 112 where there are 2 third move options for a checkmate in 2. Both options reinforce the choice of the first move square. Nonetheless, the contribution of L12H12 is relatively limited.

## H. Implementation details

Our implementation is heavily based on the implementation described in Jenner et al. (2024), and previously made available at `https://github.com/HumanCompatibleAI/leela-interp`. For the activation patching, probing, and zero ablation results, modifications were made to account for the case of more than 3 moves. For the purposes of reproducing the results, the code may be found in [link omitted for review].

**Analysis Techniques.** Our analysis builds on techniques from Jenner et al. (2024), which we detail here for completeness. For activation patching, we first run the model on the original position to get the "clean" activations. We then create a corrupted position by replacing specific moves in the game history and run the model on this corrupted position. Next, we copy activations from specific attention heads in the corrupted run into the corresponding locations in the clean run. Let $m_c$ be the correct move, $s_p$ be the patched model state, and $s_c$ be the clean model state. The log odds change $\Delta L$ of the target move is then defined as:

$$\Delta L = \log \mathrm{odds}(m_c \mid s_p) - \log \mathrm{odds}(m_c \mid s_c) \quad (1)$$

where $\log \mathrm{odds}(m_c \mid s)$ represents the logarithm of the odds that the model assigns to the correct move $m_c$ given state $s$. A negative $\Delta L$ indicates that patching reduces the model's preference for the correct move, while a positive $\Delta L$ indicates that patching increases it.

For linear probing, we extract activations from each attention head when running the model on chess positions. We then train a bilinear probe to predict the board square associated with the move of interest. The probe accuracy serves as a measure of what information is encoded by the model. The trained probe's accuracy is also compared against a random baseline.

**Puzzle generation.** Besides the dataset used by Jenner et al. (2024), we create two additional datasets. The 7-move dataset is created by starting from the 4 million puzzle dataset by filtering for puzzles with exactly 7 moves, and where the 7th move square is distinct from the other odd move squares. Additionally, as for the first dataset, we filter for puzzles that are solvable by the Leela model but not a weaker model. The generation of the alternative move dataset is described in Appendix F.

**Generating corrupted puzzles.** For the bulk of the puzzles, we rely on the implementation from Jenner et al. (2024). For the alternative move dataset, we ensure that the corrupted puzzles are viable for both branches, by applying the constraints described in Appendix D of Jenner et al. (2024) to both branches.

**Data Filtering.** To ensure reliable results, we apply several filtering criteria to the positions. For the alternative move analysis, we require the probability of each of the two first moves to be at least 0.3, and the probability of the second and third moves to be at least 0.7. Additionally, as in Jenner et al. (2024), we maintain the hardness threshold of 0.05 and forcing threshold of 0.7.

