# OpenReview forum: "Understanding the learned look-ahead behavior of chess neural networks"
_ICML.cc/2025/Conference — Submitted to ICML 2025_

### Official Review · Reviewer_EQvr · 2025-02-24

**Overall Recommendation:** 2

**Summary:**

The paper analyzes the behavior of a transformer model trained on chess games using activation patching, probing, and attention ablation, as originally proposed in [1]. The paper finds that the model considers up to the 7th future move when selecting the best next move, and its lookahead behavior is highly context-dependent (on the puzzle set). Additionally, it shows that the model considers multiple possible move sequences.

[1] Jenner et al., Evidence of Learned Look-Ahead in a Chess-Playing Neural Network, 2024

**Claims And Evidence:**

The main claims of the paper are as follows:
1. The model exhibits lookahead behavior up to the 7th move.
2. Its behavior is highly dependent on the puzzle set.
3. The model can choose an alternative move.

The first claim is well supported by Figure 3 but lacks novelty, as it is merely an extension of [1]. The second claim is not surprising, given that [1] already states, "the results of all our experiments are noticeably different on puzzles." The final claim appears to be original compared to [1], but its significance is unclear.

[1] Jenner et al., Evidence of Learned Look-Ahead in a Chess-Playing Neural Network, 2024

**Essential References Not Discussed:**

N/A

**Experimental Designs Or Analyses:**

The presentation of the experimental results could be improved, as the figures take too long to understand. The authors should introduce key concepts, such as “log odds reduction” or "residual stream," in Section 2.3 to enhance clarity.

**Methods And Evaluation Criteria:**

The proposed method is nearly identical to [1], but it is not self-contained. Regardless of its validity, the authors should have provided a more detailed explanation, as understanding it required constant reference to [1]. This lack of clarity makes it difficult to fully assess whether the method and evaluation criteria are appropriate for the problem.

[1] Jenner et al., Evidence of Learned Look-Ahead in a Chess-Playing Neural Network, 2024

**Other Comments Or Suggestions:**

I believe the authors should clearly explain how their work differs from [1]. In its current form, the paper shows very few distinctions from [1], and even the format of the figures is identical.

[1] Jenner et al., Evidence of Learned Look-Ahead in a Chess-Playing Neural Network, 2024

**Other Strengths And Weaknesses:**

N/A

**Questions For Authors:**

Please refer to the sections above.

**Relation To Broader Scientific Literature:**

The scope of this paper is limited to chess, making its connection to the broader scientific literature rather weak.

**Theoretical Claims:**

N/A

---

> ### Author Rebuttal · Authors · 2025-03-31
>
> Thank you for your review highlighting concerns about the paper's distinctiveness from Jenner et al. and its self-containedness. We would make substantial revisions in an updated version:
>
> 1. **Novelty vs. Jenner et al.**: We would revise the introduction to more clearly articulate how our work differs from and builds upon Jenner et al. Specifically, we would emphasize that we extend the analysis to the 5th and 7th future moves, reveal the network's use of distinct but consistent internal mechanisms for different move sequences, and demonstrate its ability to consider multiple possible move branches simultaneously. This would address your concern about insufficient distinction.
>
> 2. **Self-contained methods**:
>    - We would enhance the methodology section with clearer explanations of activation patching, log odds reduction, and corrupted boards, so readers don't need to constantly reference Jenner et al.
>    - We would add an explanation of log odds reduction, defining it as the decrease in log-probability that the model assigns to the correct move after patching. This would address your concern that key concepts weren't introduced in our methodology section.
>    - We would significantly expand the appendix with detailed descriptions of the implementation details, making the paper less dependent on consulting the Jenner et al. paper.
>
> 3. **Presentation improvements**:
>    - We would improve Figure 2's caption with detailed explanation of what each element represents (log odds reduction, "corrupted" label), addressing your concern that figures took too long to understand.
>    - We would enhance the clarity of puzzle set notation, making it more intuitive for readers.
>
> 4. **Broader implications**: We would strengthen the connection to broader literature in the introduction, emphasizing how the emergence of pattern-sensitive mechanisms suggests neural networks can develop generalized planning strategies applicable to novel situations, challenging the view that transformer systems merely memorize patterns without structured reasoning capabilities. This would address your concern about the limited connection to broader scientific literature.
>
> 5. **Differentiation from prior work**:
>    - We would highlight our unique contributions in the introduction and reiterate them throughout the paper to clearly distinguish our work from Jenner et al.
>    - We would reiterate specific findings about pattern-sensitive mechanisms (from the Results section) in the Conclusion, which weren't present in Jenner et al.
>
> These proposed revisions would directly address your concerns about the paper's distinctiveness and self-containedness while maintaining our core contributions.

---

> > ### Comment · Reviewer_EQvr · 2025-04-02
> >
> > Thank you for the detailed response. It would be very helpful if you could provide a more thorough explanation of the methodology, so that readers who are unfamiliar with mechanistic interpretability (including myself) can understand your paper without needing to refer to external sources. That said, I still have some concerns regarding the novelty of the paper. While some of the results are original, others appear to be straightforward extensions of prior work. I think the paper would have been stronger if it had focused more on its ability to consider multiple possible move branches simultaneously, which is not explored in prior work. As the authors responded to my question in a constructive manner, I increase my score from 1 to 2.
> >
> > Dear AC, although I personally do not find the extension of the analysis to the 5th and 7th future moves particularly novel, I recognize that it may be viewed as a meaningful contribution within the mechanistic interpretability field. I hope this context is taken into account in the evaluation of my score.

---

> > > ### Author Response · Authors · 2025-04-07
> > >
> > > Thank you for your additional feedback and the score change. You raise good points about the methodology explanation and novelty of our work.
> > >
> > > For the methodology, you're right that we should make it more self-contained. We were constrained by space limitations in the main paper, but we can definitely add clearer definitions of concepts like "log odds reduction" and "residual stream" in the main text while expanding Appendix H to provide better background for readers not familiar with mechanistic interpretability.
> > >
> > > On the novelty front, we agree that the alternative branch analysis (showing that the model considers multiple possible move sequences) is our most original contribution beyond Jenner et al. We didn't feature it as prominently in the main paper because we had a smaller dataset for this analysis (609 puzzles vs. 22k for the main analysis), so for that section we focused on the findings with stronger empirical support. We do explore this in more detail in Appendices F and G.
> > >
> > > In a revised version, we would do a better job highlighting the multiple branch analysis and what it tells us about planning in neural networks, and expanding the methodological description, especially in Appendix H.
> > >
> > > Thanks for engaging with our work and helping us improve it.

---

### Official Review · Reviewer_w8TL · 2025-03-13

**Overall Recommendation:** 4

**Summary:**

This paper builds on Jenner et al.'s work investigating the look-ahead capabilities of chess-playing neural networks, specifically the Leela Chess Zero policy network. The authors employ patching, probing, and ablation techniques to demonstrate that: 1) Chess models can consider moves up to 7 steps ahead, 2) Models evaluate alternative move sequences simultaneously, and 3) Specialized attention heads handle different aspects of chess puzzle solving tasks (e.g., L12H12 is implicated in transferring information about checkmate scenarios). The paper presents rigorous experimental evidence with detailed analyses across different puzzle types and introduces a novel notation system to classify chess positions.

**Claims And Evidence:**

The claims are well-supported by multiple empirical methods. For each main claim (look-ahead depth, simultaneous consideration of alternatives, and specialized head functions), the authors provide evidence using complementary techniques: activation patching demonstrates causal relationships, probing shows information encoding, and ablation identifies specific components responsible for behaviors. The experimental design is thorough, with careful curation of relevant puzzle subsets and controlling for confounding factors.

**Essential References Not Discussed:**

The references are sufficient, though many lack complete bibliographic information such as URLs or conference details. This is a minor issue that could be addressed in the final version.

**Experimental Designs Or Analyses:**

The experimental designs are sound and well-executed. The separation of puzzles into different sets based on move square patterns is particularly clever, allowing for disentangled analysis of how the model processes different types of positions. The combination of activation patching, probing, and ablation provides multiple lines of evidence for the claims, strengthening their validity.

**Methods And Evaluation Criteria:**

The methods are appropriate for the mechanistic interpretability questions being investigated. The authors curate subsets of the Lichess dataset specifically targeting different aspects of look-ahead behavior. Their novel notation system for reasoning about multi-step chess sequences effectively manages the combinatorial complexity of possible move sequences, enabling clearer analysis of the model's behavior across different scenarios.

**Other Comments Or Suggestions:**

* On line 84 you write "due to peculiarities of this particular model, explained in Jenner et al. 2024" - if these are not too complex, it would be good to briefly state what they are.
* On line 105 you describe your dataset. You say the solvable levels used for 3 and 5-move analysis were solvable by the Leela model. Were the additional 2.2k and 609 datasets also solvable? What percentage?
* I had difficulty understanding what the "corrupted" line in Figure 2 referred to. This could be clarified in the caption.
* On line 164, column 2 - "we particular" should be "in particular".



## Updates after rebuttal
Thank you for addressing the minor concerns raised in my review. I maintain my score of 4 as I believe this paper deserves to be accepted as a thorough extension of interesting work on the mechanistic interpretability of chess transformers; though it is not ground-breaking enough to warrant a 5.

**Other Strengths And Weaknesses:**

No additional strengths or weaknesses beyond those already mentioned.

**Questions For Authors:**

None

**Relation To Broader Scientific Literature:**

This work fits naturally within the growing field of mechanistic interpretability, a rapidly expanding subfield of AI alignment/interpretability research. It applies techniques previously applied to language models to understand strategic planning in a well-defined domain with clear evaluation metrics. The findings contribute to our understanding of how neural networks may develop planning capabilities, albeit in a toy setting.

**Theoretical Claims:**

There are no theoretical claims requiring proof in the main paper.

---

> ### Author Rebuttal · Authors · 2025-03-31
>
> Thank you for your positive review and helpful suggestions for improving clarity. We would implement all your recommended changes in a revised version of the paper:
>
> 1. **Peculiarities of model**: We would clarify in Section 2.1 that Leela originally takes in past board states in addition to the current one, and that for our analysis, we use the finetuned version from Jenner et al. (2024) that only considers the current board state. This simplifies the generation of corrupted states for activation patching while maintaining equivalent performance to the original model.
>
> 2. **Solvability of datasets**: The additional 2.2k puzzles follow the same generation principles as the original 3- and 5-move dataset, and they are also solvable by Leela. However, by design, the 609 puzzles for the alternative branch analysis require Leela to be ambivalent about two different branches (with different move choices) when picking the next move, so Leela assigns a probability of around 50% (in practice, between 30-60%) to the optimal move. Therefore, these puzzles are more difficult for the Leela model to solve.
>
> In a revised version, we would:
>    - Add text to Section 2.2 mentioning that additional details on the dataset generation and their difficulty level can be found in the appendices.
>    - Expand the "Dataset generation" section in the appendix to thoroughly explain how the new datasets are generated, and how they differ from the original dataset.
>
> 3. **"Corrupted" line clarification**:
> In a revised version, we would:
>    - Enhance the Figure 2 caption to explain that "Corrupted" indicates the square where a piece was (re)moved on the corrupted board, compared to the original board, and that higher values indicate greater importance of that square for the model's decision. This would clarify what the "corrupted" line refers to, as you requested.
>    - Clarify in the activation patching section that we slightly shift piece positions or remove non-essential pieces to create a position where the originally correct move is no longer optimal, creating a controlled comparison where most board features remain identical except for critical tactical elements.
>    - Significantly expand the "Generating corrupted puzzles" section in the appendix to explain in detail the creation of the corrupted puzzles.
>
> 4. **Grammatical issue**: We would fix "we particular" to "in particular" on page 5, line 258, as you correctly pointed out.
>
> These proposed changes would improve the clarity and completeness of our paper without altering our core findings, following your helpful suggestions.

---

### Official Review · Reviewer_DSCf · 2025-03-14

**Overall Recommendation:** 3

**Summary:**

This paper extends findings by Jenner et al (2024), which is a mechanistic interpretability paper examining how a chess network--specifically the Leela model, which has transformer architecture--"looks ahead" of game play by several moves. Specifically, the authors examine longer move sequences and possible branching behavior (i.e., alternative possible futures). They adopt a combination of patching, probing, and ablation to identify mechanisms for decision sequences. To do so, they construct a dataset (composed of three different chess puzzle sets, including the Lichess puzzles, for 3-move, 5-move, and 7-move puzzles, and cases with multiple valid move branches. By patching 3rd, 5th, and even 7th move squares, they show you can change Leela’s output — suggesting Leela considers these future moves.

**Claims And Evidence:**

The premise of this paper is exciting, though I have a few concerns with regards to experimental procedure and evaluation.

1. The patched square might be affecting behavior indirectly--via co-adapted features or general heuristics--not because the model is simulating deep future play. In Othello-GPT (Li et al., 2023), and elaborated in Nanda et al. (2023), we see that the the idea that activation interventions like patching can be compensated for by the rest of the network unless you do them carefully--usually by coordinated interventions across layers. This doesn't seem to be addressed in this experimental framework, which may be confounding the results. Specifically, they do not appear to be patching entire sequences of activations (like full attention patterns or residual streams across layers), patching across multiple layers, or compensating mechanism or propagation of the patch forward.  As probing and patching show the info is encoded and can affect output, but not necessarily that the model is deliberately using it for decision-making, it would be helpful to know if the authors did address this in some way that was unclear. In other words, "The model can encode 7-move futures" is not the same as "the model plans 7 moves ahead."

2. I didn't notice any adversarial testing or statistical robustness shown--e.g., how many patching cases don’t cause change? This is somewhat important for interpreting results.

3. The introduced notation in Section 2 is oddly complex, making it challenging to interpret results and other parts of the paper. While the notation is intended to categorize chess puzzles based on possible future moves, but reliance on sequences like 112XY or AABCD makes the results and figures harder to interpret without flipping back constantly to decode what each label means. The use of capital letters as placeholders (A, B, X, Y) for “distinct” or “arbitrary” digits is not standard and adds a symbolic burden without offering proportional clarity, and I spent some time wondering where in the alphabet the split occurs (i.e., which letters mean "distinct" versus "wildcard").

4. The binning of the dataset is follows from the notation and is reasonable, though hand-crafted, but is not evaluated statistically. The authors group puzzles using a custom binning scheme based on move destination square patterns (e.g., 112, 12345), but its unclear if there has been validation that these groupings align with meaningful model behavior or chess structure. While the bins are used extensively to interpret activation patching and attention patterns, it seems that their relevance is assumed rather than demonstrated. But this organization is the primary framework used to organize and interpret experimental findings. In other words, the binning may actually align with patterns the authors expect to find, making the results look more structured than they are. Specifically, I am concerned that there is some amount of cherry picking happening. The authors observe strong effects within a specific bins, but never test whether those effects persist across bins or on randomized groupings.

4. I also had the following questions: does the model generalize these look-ahead patterns to unseen or weird positions? Can they compare these results to a randomized or untrained version of the network?

5. The paper relies heavily on activation patching to infer causal importance of future move squares. However, the process used to generate corrupted board states--from which activations are patched--is somewhat unclear to me how it produces minimally changed board states. Is only a modification to one board placement modified? And is it validated that the assumed single patching targets only the intended feature?

**Essential References Not Discussed:**

The citations are reasonable.

**Experimental Designs Or Analyses:**

See above!

**Methods And Evaluation Criteria:**

See above

**Other Comments Or Suggestions:**

See above

**Other Strengths And Weaknesses:**

This paper combines a number of interventions into a more complete suite of interpretability tools. Some of the claims are relatively big, but depending on their responses to my above questions and concerns can be supported.

**Questions For Authors:**

see above

**Relation To Broader Scientific Literature:**

This paper introduces a novel set up for exploring future branch planning in a game setting. While it has some limitations, the findings offer interesting directions for e.g., planning through transformer-based architecture.

**Theoretical Claims:**

see above

---

> ### Author Rebuttal · Authors · 2025-03-31
>
> Thank you for your detailed review. We address each point below:
>
> 1. **Regarding indirect effects in patching**: We agree that coordinated interventions could strengthen our conclusions, though in this work we restricted our focus to patching at the layer or head level. During our initial preliminary tests, we did perform multi-head patching (possibly heads from different layers) but we did not observe notably different results from the single-head case that would warrant a deeper investigation.
>
> In a revised version, we would:
>    - Add a limitation in methodology discussion acknowledging that activation patching captures direct causal paths but may miss indirect effects distributed across multiple components
>    - Acknowledge in the conclusion that we cannot definitively determine whether the observed behavior represents true planning or sophisticated pattern matching, which would explicitly address your concern about distinguishing between encoding information and using it for planning
>
> 2. **Statistical robustness**: All puzzle sets shown contained at least 50 puzzles, which we deemed sufficient for statistical reliability. Exceptions are two puzzle sets (with 36 and 49 examples) for the alternative branch analysis, where dataset constraints limited our samples. Our appendix includes results for all qualifying puzzle sets, including those where activation patching shows no meaningful effect on log odds reduction (e.g., set 122 in Figure 8), demonstrating we didn't cherry-pick only positive results.
>
> In a revised version, we would explicitly state that we analyzed all puzzle sets with at least 50 examples to ensure statistical reliability, covering the full range of move patterns in our dataset.
>
> 3. **Notation and binning**: We appreciate your feedback on the notation. While we developed it to manage the combinatorial complexity of move sequences, we recognize it could be clearer. Our notation followed the common convention that initial alphabet letters (A, B, C, ...) denote constants, while final letters (X, Y, Z) denote variables. We understand that our use of capital letters and the additional usage of M and N might have contributed to the confusion. In practice, we only use the letters A, B, C, D, X, Y, and Z. The letter M was chosen as shorthand for "mate", and N for "non-(check)mate".
>
> Our binning approach stems from two key considerations:
>    - As patching is applied to board squares associated with residual stream dimensions, different puzzle sets naturally show qualitatively different behaviors. For example, in set 112, patching cannot distinguish between first and second moves (same square), while in set 123, all moves use distinct squares.
>    - Our preliminary experiments revealed marked different patching behavior for checkmate versus non-checkmate positions, leading to our M/N prefixes.
>
> We note that some puzzles within sets show behaviors deviating from the typical pattern, suggesting potential additional meaningful categorizations. For instance, in set M112 (Figure 5), most puzzles respond strongly to ablation, but about a quarter present minimal changes.
>
> In a revised version, we would restructure the puzzle set notation section with a clearer explanation of how we categorize puzzles based on the pattern of squares pieces move to.
>
> 4. **Generalization to unseen positions**: We use an untrained chess model as a baseline for probing results (dashed lines in Figure 3), showing that Leela encodes future move information that random models don't. In Appendix G, we tested handcrafted positions with two possible checkmates in 2 moves - a scenario absent from the 4 million Lichess puzzles. Despite being unlikely to appear in training data, attention head L12H12 still moves information "backward in time" as expected, suggesting generalization of look-ahead behavior.
>
> 5. **Corrupted board state generation**: Our corruption process minimally modifies the board state by changing a single piece's position, which changes the optimal move while preserving most board features. We verify that this process mainly affects the intended feature by showing localized changes in attention patterns.
>
> In a revised version, we would:
>    - Clarify that we slightly shift piece positions or remove non-essential pieces to create positions where the originally correct move is no longer optimal, creating controlled comparisons where most board features remain identical except for critical tactical elements
>    - Expand the appendix explanation of corrupted puzzle generation methodology
>
> We believe these proposed changes would directly address your methodological concerns while acknowledging the limitations of our approach. Our multi-faceted analysis using complementary techniques (patching, probing, and ablation) helps mitigate the limitations of any single approach, allowing us to build a more comprehensive understanding of the model's look-ahead behavior.

---

### Official Review · Reviewer_LpmU · 2025-03-19

**Overall Recommendation:** 2

**Summary:**

The authors use an existing technique for examining chess model internal states to expand the analysis of chess games to more complex positions.

**Claims And Evidence:**

This paper has a common issue for interp papers, the authors don't make strong claims.

Of the three key contributions two (first and last) are not surprising or relevent without generazliation to other domains, while the middle one is an hypothesis without complete support.

**Essential References Not Discussed:**

No

**Experimental Designs Or Analyses:**

I don't find these methods very convincing, but as they rely on previous papers that is not the question for this review.

The experiments appear to be a direct expansion on the previous work, but without presenting any new theoretical results. This might be relevant to a more specialized community, but the details of Leela's look ahead are not enough to support the claims of larger insights.

**Methods And Evaluation Criteria:**

The methods make sense, but the dataset is similar to the previous work so the results are accordingly narrow.

**Other Comments Or Suggestions:**

I apologize for my short review, due to circumstances my verbosity is hampered

**Other Strengths And Weaknesses:**

The expansion of on the previous Jenner paper is good, and maybe will lead to deeper insights in time

**Questions For Authors:**

No

**Relation To Broader Scientific Literature:**

I think there is significant merit in understanding the inner workings of NNs, but this paper does not contribute to the broader understanding. Another venue might be considered by the authors

**Theoretical Claims:**

They present no theorems, and the theory used is from other papers (Jenner et al., 2024).

---

> ### Author Rebuttal · Authors · 2025-03-31
>
> Thank you for your review. We appreciate your feedback on our paper's contributions and their broader relevance.
> We wish to clarify that our work makes several substantive contributions beyond Jenner et al. (2024):
>
> 1. While Jenner et al. showed evidence of look-ahead to the 3rd move, we demonstrate that the network can process information up to the 7th move and analyze the mechanisms involved, representing a significant extension of the original work's scope.
> 2. Our analysis of the model's ability to consider multiple possible move sequences (alternative branches) is entirely novel and not covered in Jenner et al. This finding has important implications for understanding how neural networks can develop tree search-like capabilities without explicit programming.
> 3. Our identification of specialized attention heads for different types of positions (e.g., checkmate vs. non-checkmate scenarios) provides new mechanistic insights into how neural networks develop specialized components for different strategic contexts.
>
> Regarding broader relevance, our findings about emergent planning capabilities in neural networks extend beyond chess. They contribute to our understanding of how models can learn to simulate future states and alternative possibilities through training—a capability relevant to autonomous systems, robotics, and any AI that needs to plan multiple steps ahead in complex environments.
>
> We understand your concern that our paper may not make strong claims beyond Jenner et al., and that the generalization to other domains is limited. We would address these issues in a revised version of the paper as follows:
>
> 1. **Strengthen our claims with additional supporting evidence**:
>    - In the introduction, we would clarify our contributions beyond Jenner et al., emphasizing that we not only extend to higher move counts but also identify pattern-sensitive mechanisms that operate across different time horizons.
>    - We would add a section highlighting that our observed attention patterns represent meaningful structure rather than cherry-picked examples, providing evidence that our findings represent general mechanisms rather than isolated observations, strengthening the validity of our claims.
>
> 2. **Improve generalization**:
>    - We would emphasize more clearly that the specific patterns attention heads respond to appear to be time-insensitive, suggesting the model has learned general pattern-matching mechanisms across time rather than timing-specific heuristics. This directly addresses your concern about generalization by showing that the patterns we've identified are not specific to particular time steps but represent general strategies.
>    - We would revise the conclusion to emphasize the claim of pattern-sensitive mechanisms that could generalize beyond chess, making our contribution more broadly relevant to AI reasoning.
>
> 3. **Clarify theoretical importance**:
>    - We would enhance the introduction to connect the emergence of pattern-sensitive mechanisms to broader questions about how neural networks can develop generalized planning strategies applicable to novel situations.
>    - We would strengthen the conclusion about challenging "simplistic views of neural networks as merely statistical pattern matchers," highlighting the theoretical significance of our findings beyond the narrow domain of chess.
>
> While our work builds on Jenner et al., we believe these proposed revisions would highlight our novel contributions to understanding how neural networks develop sophisticated look-ahead capabilities that could inform AI research beyond chess.

---

### Decision · Program_Chairs · 2025-05-01

**Decision:**

Reject

**Comment:**

This paper investigates the look-ahead behavior of the Leela Chess Zero policy network. The authors use several explanation techniques (activation patching, probing, and ablation) to analyze how models understand future game states. The results demonstrate that models can process information up to seven moves ahead. These provide new insights for reasoning and interpretability.

One reviewer supports clear acceptance, recognizing solid and thorough experiments. However, other reviewers raised concerns about the novelty and the limited contribution compared to previous work. After the discussion, two reviewers remained unconvinced. Overall, I recommend rejection and suggest the authors improve the presentation for future submission.